# AUTOENCODING HYPERBOLIC REPRESENTATION FOR ADVERSARIAL GENERATION

## ABSTRACT

With the recent advance of geometric deep learning, neural networks have been extensively used for data in non-Euclidean domains. In particular, hyperbolic neural networks have proved successful in processing hierarchical information of data. However, many hyperbolic neural networks are numerically unstable during training, which precludes using complex architectures. This crucial problem makes it difficult to build hyperbolic generative models for real and complex data. In this work, we propose a hyperbolic generative network in which we design novel architecture and layers to improve stability in training. Our proposed network contains three parts: first, a hyperbolic autoencoder (AE) that produces hyperbolic embedding for input data; second, a hyperbolic generative adversarial network (GAN) for generating the hyperbolic latent embedding of the AE from simple noise; third, a generator that inherits the decoder from the AE and the generator from the GAN. We call this network the hyperbolic AE-GAN, or HAE-GAN for short. The architecture of HAEGAN fosters expressive representation in the hyperbolic space, and the specific design of layers ensures numerical stability. Experiments show that HAEGAN is able to generate complex data with state-of-the-art structure-related performance.

## 1 INTRODUCTION

High-dimensional data often show an underlying geometric structure, which cannot be easily captured by neural networks designed for Euclidean spaces. Recently, there is intense interest in learning good representation for hierarchical data, for which the most natural underlying geometry is hyperbolic. A hyperbolic space is a Riemannian manifold with a constant negative curvature (Anderson, 2006). The exponential growth of the radius of the hyperbolic space provides high capacity, which makes it particularly suitable for modeling tree-like hierarchical structures. Hyperbolic representation has been successfully applied to, for instance, social network data in product recommendation (Wang et al., 2019), molecular data in drug discovery (Yu et al., 2020; Wu et al., 2021), and skeletal data in action recognition (Peng et al., 2020).

Many recent works (Ganea et al., 2018; Shimizu et al., 2021; Chen et al., 2021) have successfully designed hyperbolic neural operations. These operations have been used in generative models for generating samples in the hyperbolic space. For instance, several recent works (Nagano et al., 2019; Mathieu et al., 2019; Dai et al., 2021b) have built hyperbolic variational autoencoders (VAE) (Kingma & Welling, 2014). On the other hand, Lazcano et al. (2021) have generalized generative adversarial networks (GAN) (Goodfellow et al., 2014; Arjovsky et al., 2017) to the hyperbolic space. However, the above hyperbolic generative models are known to suffer from gradient explosion when the networks are deep. In order to build hyperbolic networks that can generate real data, it is desired to have a framework that has both representation power and numerical stability.

To this end, we design a novel hybrid model which learns complex structures and hyperbolic embeddings from data, and then generates examples by sampling from random noises in the hyperbolic space. Altogether, our model contains three parts: first, we use a hyperbolic autoencoder (AE) to learn the embedding of training data in the latent hyperbolic space; second, we use a hyperbolic GAN to learn generating the latent hyperbolic distribution by passing a wrapped normal noise through the generator; third, we generate samples by applying sequentially the generator of the GAN and the decoder of the AE. We name our model as Hyperbolic AE-GAN, or HAEGAN for short. The advan-

tage of this architecture is twofold: first, it enjoys expressivity since the noise goes through both the layers of the generator and the decoder; second, it allows flexible design of the AE according to the type of input data, which does not affect the sampling power of GAN. In addition, HAEGAN avoids the complicated form of ELBO in hyperbolic VAE, which is one source of numerical instability. We highlight the main contributions of this paper as follows:

- HAEGAN is a novel hybrid AE-GAN framework for learning hyperbolic distributions that aims for both expressivity and numerical stability.
- We validate the Wasserstein GAN formulation in HAEGAN, especially the way of sampling from the geodesic connecting a real sample and a generated sample..
- We design a novel concatenation layer in the hyperbolic space. We extensively investigate its numerical stability via theoretical and experimental comparisons.
- In the experiments part, we illustrate that HAEGAN is not only able to faithfully generate synthetic hyperbolic data, but also able to generate real data with sound quality. In particular, we consider the molecular generation task and show that HAEGAN achieves state-of-the-art performance, especially in metrics related to structural properties.

## 2 BACKGROUND IN HYPERBOLIC NEURAL NETWORKS

### 2.1 HYPERBOLIC GEOMETRY

Hyperbolic geometry is a special kind of Riemannian geometry with a constant negative curvature (Cannon et al., 1997; Anderson, 2006). To extract hyperbolic representations, it is necessary to choose a "model", or coordinate system, for the hyperbolic space. Popular choices include the Poincaré ball model and the Lorentz model, where the latter is found to be numerically more stable (Nickel & Kiela, 2018). We work with the Lorentz model $\mathbb{L}_K^n = (\mathcal{L}, \mathfrak{g})$ with a constant negative curvature $K$, which is an $n$-dimensional manifold $\mathcal{L}$ embedded in the $(n+1)$-dimensional Minkowski space, together with the Riemannian metric tensor $\mathfrak{g} = \text{diag}([-1, \mathbf{1}_n^\top])$, where $\mathbf{1}_n$ denotes the $n$-dimensional vector whose entries are all 1's. Every point in $\mathbb{L}_K^n$ is represented by $\boldsymbol{x} = [x_t, \boldsymbol{x}_s^\top]^\top, x_t > 0, \boldsymbol{x}_s \in \mathbb{R}^n$, and satisfies $\langle \boldsymbol{x}, \boldsymbol{x} \rangle_\mathcal{L} = 1/K$, where $\langle \cdot, \cdot \rangle_\mathcal{L}$ is the Lorentz inner product induced by $\mathfrak{g}^K : \langle \boldsymbol{x}, \boldsymbol{y} \rangle_\mathcal{L} := \boldsymbol{x}^\top \mathfrak{g} \boldsymbol{y} = -x_t y_t + \boldsymbol{x}_s^\top \boldsymbol{y}_s, \ \boldsymbol{x}, \boldsymbol{y} \in \mathbb{L}_K^n$. In the rest of the paper, we will refer to $x_t$ as the "time component" and $\boldsymbol{x}_s$ as the "spatial component" . In the following, we describe some notations. Extensive details are provided in Appendix A.

**Notation** We use $d_\mathcal{L}(\boldsymbol{x}, \boldsymbol{y})$ to denote the length of a geodesic ("distance" along the manifold) connecting $\boldsymbol{x}, \boldsymbol{y} \in \mathbb{L}_K^n$. For each point $\boldsymbol{x} \in \mathbb{L}_K^n$, the tangent space at $\boldsymbol{x}$ is denoted by $\mathcal{T}_{\boldsymbol{x}} \mathbb{L}_K^n$. The norm $\|\cdot\|_\mathcal{L} = \sqrt{\langle \cdot, \cdot \rangle_\mathcal{L}}$. For $\boldsymbol{x}, \boldsymbol{y} \in \mathbb{L}_K^n$ and $\boldsymbol{v} \in \mathcal{T}_{\boldsymbol{x}} \mathbb{L}_K^n$, we use $\exp_{\boldsymbol{x}}^K(\boldsymbol{v})$ to denote the exponential map of $\boldsymbol{v}$ at $\boldsymbol{x}$; on the other hand, we use $\log_{\boldsymbol{x}}^K : \mathbb{L}_K^n \to \mathcal{T}_{\boldsymbol{x}} \mathbb{L}_K^n$ to denote the logarithmic map such that $\log_{\boldsymbol{x}}^K(\exp_{\boldsymbol{x}}^K(\boldsymbol{v})) = \boldsymbol{v}$. For two points $\boldsymbol{x}, \boldsymbol{y} \in \mathbb{L}_K^n$, we use $\text{PT}_{\boldsymbol{x} \to \boldsymbol{y}}^K$ to denote the parallel transport map which "transports" a vector from $\mathcal{T}_{\boldsymbol{x}} \mathbb{L}_K^n$ to $\mathcal{T}_{\boldsymbol{y}} \mathbb{L}_K^n$ along the geodesic from $\boldsymbol{x}$ to $\boldsymbol{y}$.

### 2.2 FULLY HYPERBOLIC LAYERS

One way to define hyperbolic neural operations is to use the tangent space, which is Euclidean. However, working with the tangent space requires taking exponential and logarithmic maps, which cause numerical instability. Moreover, tangent spaces are only local estimates of the hyperbolic spaces, but neural network operations are usually not local. Since generative networks have complex structures, we want to avoid using the tangent space whenever possible. The following hyperbolic layers take a "fully hyperbolic" approach and perform operations directly in hyperbolic spaces.

The most fundamental hyperbolic neural layer, the *hyperbolic linear layer* (Chen et al., 2021), is a trainable "linear transformation" that maps from $\mathbb{L}_K^n$ to $\mathbb{L}_K^m$. We remark that "linear" is used to analogize the Euclidean counterpart, which contains activation, bias and normalization. In general, for an input $\boldsymbol{x} \in \mathbb{L}_K^n$, the hyperbolic linear layer outputs

$$\boldsymbol{y} = \text{HLinear}_{n,m}(\boldsymbol{x}) = \begin{bmatrix} \sqrt{\|h(\boldsymbol{W}\boldsymbol{x}, \boldsymbol{v})\|^2 - 1/K} \\ h(\boldsymbol{W}\boldsymbol{x}, \boldsymbol{v}) \end{bmatrix}. \tag{1}$$

Here $h(\boldsymbol{W}\boldsymbol{x}, \boldsymbol{v}) = \frac{\lambda\sigma(\boldsymbol{v}^\top\boldsymbol{x}+b')}{\|\boldsymbol{W}\tau(\boldsymbol{x})+\boldsymbol{b}\|}(\boldsymbol{W}\tau(\boldsymbol{x}) + \boldsymbol{b})$, where $\boldsymbol{v} \in \mathbb{R}^{n+1}$ and $\boldsymbol{W} \in \mathbb{R}^{m\times(n+1)}$ are trainable weights, $\boldsymbol{b}$ and $b'$ are trainable biases, $\sigma$ is the sigmoid function, $\tau$ is the activation function, and the trainable parameter $\lambda > 0$ scales the range.

While a map between hyperbolic spaces such as HLinear is used most frequently in hyperbolic networks, it is also often necessary to output Euclidean features. A numerically stable way of obtaining Euclidean output features is via the *hyperbolic centroid distance layer* (Liu et al., 2019), which maps points from $\mathbb{L}_K^n$ to $\mathbb{R}^m$. Given an input $\boldsymbol{x} \in \mathbb{L}_K^n$, it first initializes $m$ trainable centroids $\{\boldsymbol{c}_i\}_{i=1}^m \subset \mathbb{L}_K^n$, then produces a vector of distances

$$\boldsymbol{y} = \text{HCDist}_{n,m}(\boldsymbol{x}) = [d_{\mathcal{L}}(\boldsymbol{x}, \boldsymbol{c}_1)\cdots d_{\mathcal{L}}(\boldsymbol{x}, \boldsymbol{c}_m)]^\top. \tag{2}$$

We introduce more hyperbolic neural layers used in our model in Appendix A.2.

## 3 HYPERBOLIC AUTO-ENCODER GENERATIVE ADVERSARIAL NETWORKS

### 3.1 HYPERBOLIC GAN

With the hyperbolic neural layers defined in §2.2, it is not difficult to define a hyperbolic GAN whose generator and critic contain hyperbolic linear layers. The critic also contains a centroid distance layer so that its output is a one-dimensional score. To produce generated samples, we sample $\boldsymbol{z}^{(0)}$ from $\mathcal{G}(\boldsymbol{o}, \boldsymbol{I})$, the wrapped normal distribution (Nagano et al., 2019) and then pass it through the generator. We follow the Wasserstein gradient penalty (WGAN-GP) framework (Gulrajani et al., 2017) to foster easy and stable training. The loss function, adapted from the Euclidean WGAN-GP, is

$$L_{\text{WGAN}} = \mathbb{E}_{\tilde{\boldsymbol{x}}\sim\mathbb{P}_g}[D(\tilde{\boldsymbol{x}})] - \mathbb{E}_{\boldsymbol{x}\sim\mathbb{P}_r}[D(\boldsymbol{x})] + \lambda\mathbb{E}_{\hat{\boldsymbol{x}}\sim\mathbb{P}_{\hat{x}}}\left[\left(\|\nabla D(\hat{\boldsymbol{x}})\|_{\mathcal{L}} - 1\right)^2\right], \tag{3}$$

where $D$ is the critic, $\nabla D(\hat{\boldsymbol{x}})$ is the Riemannian gradient of $D(\boldsymbol{x})$ at $\hat{\boldsymbol{x}}$, $\mathbb{P}_g$ is the generator distribution and $\mathbb{P}_r$ is the data distribution. Most importantly, $\mathbb{P}_{\hat{x}}$ samples uniformly along the geodesic between pairs of points sampled from $\mathbb{P}_g$ and $\mathbb{P}_r$ instead of a linear interpolation. This manner of sampling is validated in the following proposition, the proof of which can be found in §B.

**Proposition 3.1.** *Let $\mathbb{P}_r$ and $\mathbb{P}_g$ be two distributions in $\mathbb{L}_K^n$ and $f^*$ be an optimal solution of $\max_{\|f\|_L \leq 1} \mathbb{E}_{\boldsymbol{y}\sim\mathbb{P}_r}[f(\boldsymbol{y})] - \mathbb{E}_{\boldsymbol{x}\sim\mathbb{P}_g}[f(\boldsymbol{x})]$ where $\|\cdot\|_L$ is the Lipschitz norm. Let $\pi$ be the optimal coupling between $\mathbb{P}_r$ and $\mathbb{P}_g$ that minimizes $W(\mathbb{P}_r, \mathbb{P}_g) = \inf_{\pi\in\Pi(\mathbb{P}_r,\mathbb{P}_g)} \mathbb{E}_{x,y\sim\pi}[d_{\mathcal{L}}(\boldsymbol{x}, \boldsymbol{y})]$, where $\Pi(\mathbb{P}_r, \mathbb{P}_g)$ is the set of joint distributions $\pi(\boldsymbol{x}, \boldsymbol{y})$ whose marginals are $\mathbb{P}_r$ and $\mathbb{P}_g$, respectively. Let $\boldsymbol{x}_t = \gamma(t), 0 \leq t \leq 1$ be the geodesic between $\boldsymbol{x}$ and $\boldsymbol{y}$, such that $\gamma(0) = \boldsymbol{x}, \gamma(1) = \boldsymbol{y}, \gamma'(t) = \boldsymbol{v}_t \in \mathcal{T}\mathbb{L}_K^n, \|\boldsymbol{v}_t\|_{\mathcal{L}} = d_{\mathcal{L}}(\boldsymbol{x}, \boldsymbol{y})$. If $f^*$ is differentiable and $\pi(\boldsymbol{x} = \boldsymbol{y}) = 0$, then it holds that*

$$\mathbb{P}_{(\boldsymbol{x},\boldsymbol{y})\sim\pi}\left[\nabla f^*(\boldsymbol{x}_t) = \frac{\boldsymbol{v}_t}{d_{\mathcal{L}}(\boldsymbol{x}, \boldsymbol{y})}\right] = 1. \tag{4}$$

The WGAN-GP formulation of the hyperbolic GAN is capable of sampling distributions in low-dimensional hyperbolic spaces. We illustrate learning 2D distributions in §B.

### 3.2 ARCHITECTURE OF HAEGAN

Although a hyperbolic GAN defined above can faithfully generate hyperbolic distributions , its training is difficult. Specifically, numerical instability will be observed when we incorporate complex network architectures into the generator and the critic. To this end, we design our HAEGAN model to contain both a hyperbolic AE and a hyperbolic GAN. First, we train a hyperbolic AE and use the encoder to embed the dataset into a latent hyperbolic space. Then, we use our hyperbolic GAN to learn the latent distribution of the embedded data. Finally, we sample hyperbolic embeddings using the generator and use the decoder to get samples in the original space. An illustration of HAEGAN is shown in Figure 1.

In addition to expressivity, HAEGAN enjoys flexibility in choosing the AE for embedding the hyperbolic distribution. If the original dataset is not readily presented in the hyperbolic domain, the hyperbolic AE can also learn the hyperbolic representation. In this case, we adopt the hyperbolic

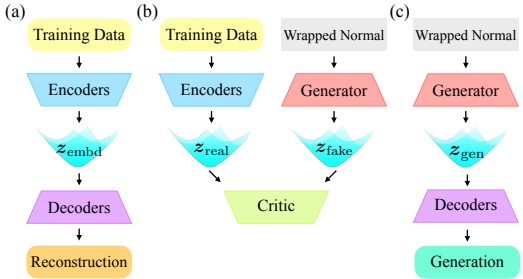

Figure 1: Overview of HAEGAN. (a) The hyperbolic AE. (b) The hyperbolic GAN for generating the latent embeddings. The encoders in (b) are identical to (a). (c) The process for sampling molecules. The generator in (c) is identical to (b) and the decoders in (c) are identical to (a).

embedding operation (Nagano et al., 2019) from Euclidean to the hyperbolic space. We review the details of this embedding operation in Appendix A.3.

As a sanity check, we train a HAEGAN with the MNIST dataset (LeCun et al., 2010) and present some generated samples in Figure 2. It is clear that HAEGAN can faithfully generate synthetic examples. We describe the details and perform a quantitative comparison with other hyperbolic models regarding this task in Appendix C.

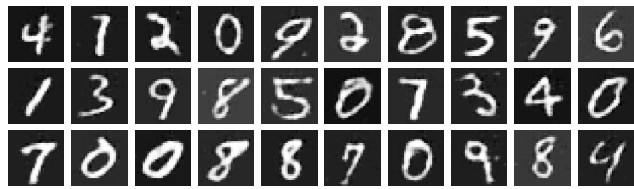

Figure 2: Samples generated from the HAEGAN trained on MNIST.

## 4 LORENTZ CONCATENATION

### 4.1 MOTIVATION AND DEFINITION

Concatenation and split are essential operations in neural networks for feature combination, parallel computation, etc. For data with complex structures, the decoder of the hyperbolic AE in HAEGAN will need to produce parts of features and then combine them, where we need to perform concatenation. However, there is no obvious way of doing concatenation in the hyperbolic space. Shimizu et al. (2021) proposed Poincaré $\beta$-concatenation and $\beta$-split in the Poincaré model. Specifically, they first use the logarithmic map to lift hyperbolic points to the tangent plane of the origin, then perform Euclidean concatenation and split in this tangent space, and finally apply $\beta$ regularization and apply the exponential map to bring it back to the Poincaré ball.

Since we use the Lorentz model, the above operations are not useful and we need to define concatenation and split in the Lorentz space. One could define operations in the tangent space similarly to the Poincaré $\beta$-concatenation and $\beta$-split. More specifically, if we want to concatenate the input vectors $\{x_i\}_{i=1}^N$ where each $x_i \in \mathbb{L}_K^{n_i}$, we could follow a "Lorentz tangent concatenation": first lift each $x_i$ to $v_i = \log_o^K(x_i) = \begin{bmatrix} v_{i_t} \\ v_{i_s} \end{bmatrix} \in \mathbb{R}^{n_i+1}$, and then perform the Euclidean concatenation to get $v := \begin{bmatrix} 0, v_{1_s}^\top, \ldots, v_{N_s}^\top \end{bmatrix}^\top$. Finally, we would get $y = \exp_o^K(v)$ as a concatenated vector in the hyperlolic space. We denote $y = \mathrm{HTCat}(\{x_i\}_{i=1}^N)$. Similarly, we could perform the "Lorentz tangent split" on an input $x_i \in \mathbb{L}_K^n$ with split sub-dimensions $\sum_{i=1}^N n_i = n$ to get $v = \log_o^K(x) = \begin{bmatrix} 0, v_{1_s}^\top \in \mathbb{R}^{n_1}, \ldots, v_{N_s}^\top \in \mathbb{R}^{n_N} \end{bmatrix}^\top$, $v_i = \begin{bmatrix} 0 \\ v_{i_s} \end{bmatrix} \in \mathcal{T}_o\mathbb{L}_K^{n_i}$, and the split vectors $y_i = \exp_o^K(v_i)$ successively.

Unfortunately, there are two problems with both the Lorentz tangent concatenation and the Lorentz tangent split. First, they are not "regularized", which means that the norm of the spatial component will increase after concatenation, and decrease after split. This will make the hidden embeddings numerically unstable. This problem could be partially solved by adding a hyperbolic linear layer after each concatenation and split, similarly to Ganea et al. (2018), so that we have a trainable scaling factor $\lambda$ to regularize the norm of the output. The second and more important problem is that if we use the Lorentz tangent concatenation and split in a deep neural network, there would be too many exponential and logarithmic maps. It on one hand suffers from severe precision issue due to the inaccurate float representation (Yu & De Sa, 2019; 2021), and on the other hand easily suffers from gradient explosion. Moreover, the tangent space is chosen at $\boldsymbol{o}$. If the points to concatenate are not close to $\boldsymbol{o}$, their hyperbolic relation may not be captured very well. Therefore, we abandon the use of the tangent space and propose more direct and numerically stable operations, which we call the "Lorentz direct concatenation and split", defined as follows.

Given the input vectors $\{\boldsymbol{x}_i\}_{i=1}^N$ where each $\boldsymbol{x}_i \in \mathbb{L}_K^{n_i}$ and $M = \sum_{i=1}^N n_i$, the Lorentz direct concatenation of $\{\boldsymbol{x}_i\}_{i=1}^N$ is defined to be a vector $\boldsymbol{y} \in \mathbb{L}_k^M$ given by

$$\boldsymbol{y} = \mathrm{HCat}(\{\boldsymbol{x}_i\}_{i=1}^N) = \left[ \sqrt{\sum_{i=1}^N x_{i_t}^2 + (N-1)/K}, \boldsymbol{x}_{1_s}^\top, \cdots, \boldsymbol{x}_{N_s}^\top \right]^\top. \tag{5}$$

Note that each $\boldsymbol{x}_{i_s}$ is the spatial component of $\boldsymbol{x}_i$. If we consider $\boldsymbol{x}_i \in \mathbb{L}_K^{n_i}$ as a point in $\mathbb{R}^{n_i+1}$, the projection of $\boldsymbol{x}_i$ onto the Euclidean subspace $\{0\} \times \mathbb{R}^n$, or the closest point there, is $\boldsymbol{x}_{i_s}$. The Lorentz direct concatenation can thus be considered as Euclidean concatenation of projections, where the Euclidean concatenated point is mapped back to $\mathbb{L}_K^M$ by the inverse map of the projection. We remark that this concatenation directly inherits from the Lorentz model.

We also define the Lorentz split for completeness, though our main focus is on concatenation: Given an input $\boldsymbol{x} \in \mathbb{L}_K^n$, the Lorentz direct split of $\boldsymbol{x}$, with sub-dimensions $n_1, \cdots, n_N$ where $\sum_{i=1}^N n_i = n$, will be $\{\boldsymbol{y}_i\}_{i=1}^N$, where each $\boldsymbol{y}_i \in \mathbb{L}_K^{n_i}$ is given by first splitting $\boldsymbol{x}$ as $\boldsymbol{x} = \left[x_t, \boldsymbol{y}_{1_s}^\top, \cdots, \boldsymbol{y}_{N_s}^\top\right]^\top$, and then calculating the corresponding time dimension as $\boldsymbol{y}_i = \left[ \begin{smallmatrix} \sqrt{\|\boldsymbol{y}_{i_s}\|^2 - 1/K} \\ \boldsymbol{y}_{i_s} \end{smallmatrix} \right]$.

## 4.2 ADVANTAGE OF LORENTZ DIRECT CONCATENATION

We show the advantage of direct concatenation in this section. Firstly, We state the following theoretical result regarding the exploding gradient of the Lorentz tangent concatenation.

**Theorem 4.1.** *Let* $\{\boldsymbol{x}_i\}_{i=1}^N$, *where* $\boldsymbol{x}_i \in \mathbb{L}_K^{n_i}$, *denote the input features. Let* $\boldsymbol{y} = \mathrm{HCat}(\{\boldsymbol{x}_i\}_{i=1}^N)$ *denote the output of the Lorentz direct concatenation and* $\boldsymbol{z} = \mathrm{HTCat}(\{\boldsymbol{x}_i\}_{i=1}^N)$ *denote the output of the Lorentz tangent concatenation. Fix* $j \in \{1, \cdots, N\}$. *The following results hold:*

1. *For any* $\{\boldsymbol{x}_i\}_{i=1}^N$ *and any entry* $y^*$ *of* $\boldsymbol{y}$, $\|\partial y^*/\partial \boldsymbol{x}_{j_s}|_{\boldsymbol{x}_1, \cdots, \boldsymbol{x}_N}\| \leq 1$.
2. *For any* $M > 0$, *there exist* $\{\boldsymbol{x}_i\}_{i=1}^N$ *and an entry* $z^*$ *of* $\boldsymbol{z}$ *for which* $\|\partial z^*/\partial \boldsymbol{x}_{j_s}|_{\boldsymbol{x}_1, \cdots, \boldsymbol{x}_N}\| \geq M$.

This theorem shows that while the Lorentz direct concatenation has bounded gradients, there is no control on the gradients of Lorentz tangent concatenation. The proof can be found in Appendix D.1 and we give a simple numerical validation in Appendix D.2.

We remark that in addition to ensuring bounded gradients, discarding exponential and logarithmic maps also makes sure that the concatenation operation does not suffer from inaccurate float representations (Yu & De Sa, 2019; 2021). There are two additional benefits: first, the direct concatenation is less complex and thus more efficient than the tangent concatenation; second, the operation does not contain exponential functions and is thus GPU scalable (Choudhary & Reddy, 2022).

Next, we design the following simple experiment to show the advantage of our Lorentz direct concatenation over the Lorentz tangent concatenation when they are used in simple neural networks. The hyperbolic neural network in this simple experiment consists of a cascading of $L$ blocks . A $d$-dimensional input is fed into two different hyperbolic linear layers, whose outputs are then concatenated by the Lorentz direct concatenation and the Lorentz tangent concatenation, respectively. Then, the concatenated output further goes through another hyperbolic linear layer whose output is

again $d$-dimensional. Specifically, for $l = 0, \cdots, L - 1$,

$$
\begin{aligned}
\boldsymbol{h}_1^{(l)} &= \text{Hlinear}_{d,d}(\boldsymbol{x}^{(l)}), & \boldsymbol{h}_2^{(l)} &= \text{Hlinear}_{d,d}(\boldsymbol{x}^{(l)}); \\
\boldsymbol{h}^{(l)} &= \text{HCat}(\boldsymbol{h}_1^{(l)}, \boldsymbol{h}_2^{(l)}); & \boldsymbol{x}^{(l+1)} &= \text{Hlinear}_{2 \times d, d}(\boldsymbol{h}^{(l)}).
\end{aligned}
\tag{6}
$$

In our test, we take $d = 64$. We sample input and output data from two wrapped normal distributions with different means (input: origin $\boldsymbol{o}$, output: $\text{E2H}(\mathbf{1}_{64})$) and variances (input: $\text{diag}(\mathbf{1}_{64})$, output: $3 \times \text{diag}(\mathbf{1}_{64})$). Taking the input as $\boldsymbol{x}^{(0)}$, we fit $\boldsymbol{x}^{(L)}$ to the output data. We record the average gradient norm of the three hyperbolic linear layers in each block. The results for $L = 64$ blocks and $L = 128$ blocks are shown in Figure 3. Clearly, for the first 20 blocks, the Lorentz tangent concatenation leads to significantly larger gradient norms. This difference in norms is clearer when the network is deeper. The gradients from the Lorentz direct concatenation are much more stable.

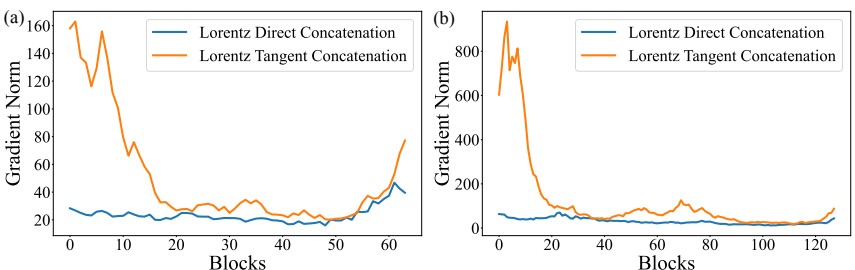

Figure 3: Average gradient norm of the each block in training. (a) 64 blocks. (b) 128 blocks.

Finally, we remark that Lorentz direct concatenation is numerically stable in practice. In the generation experiments which we introduce in the next section, if we used the Lorentz tangent concatenation, the gradients would explode very rapidly (produce NaN), even during the first epoch.

More analysis about hyperbolic concatenations, particularly regarding their impact on hyperbolic distances and their stability, can be found in Appendix D.3.

## 5 EXPERIMENTS

We perform the following two experiments: random tree generation and molecular generation.

### 5.1 RANDOM TREE GENERATION

Recent studies (Boguñá et al., 2010; Krioukov et al., 2010; Sala et al., 2018; Sonthalia & Gilbert, 2020) have found that hyperbolic spaces are suitable for tree-like graphs. We first perform an experiment in which we generate random trees. We compare the performance of HAEGAN with other hyperbolic models. In this experiment, the AE consists of a tree encoder and a tree decoder. The structure of the AE is explained in detail in Appendix E. We remark that both the tree encoder and the tree decoder are also used in the molecular generation experiment.

**Dataset** Our dataset consists of 500 randomly generated trees. Each tree is created by converting a uniformly random Prüfer sequence (Prüfer, 1918). The number of nodes in each tree is uniformly sampled from $[20, 50]$. The dataset is randomly split into 400 for training and 100 for testing.

**Baselines and Ablations** We compare HAEGAN with the following baseline hyperbolic generation methods. HGAN, where we only use a hyperbolic Wasserstein GAN without the AE structure, where the tree decoder as the generator and the tree decoder as the critic. HVAE-w and HVAE-r, where we use the same AE but follow the ELBO loss function used by Mathieu et al. (2019) instead of having a GAN ("w" and "r" refer to using wrapped and Riemannian normal distributions, respectively). Although we mainly focus on hyperbolic methods, we also compare with the following

Euclidean generation methods: GraphRNN (You et al., 2018b) and AEGAN. The latter has the same architecture as HAEGAN but all layers and operations are in the Euclidean space.

As discussed in previous sections, our default choice in HAEGAN is to use Lorentz *direct* Concatenation and *fully* hyperbolic linear layers (Chen et al., 2021). We also consider the following ablations of HAEGAN: HAEGAN-H, where the fully hyperbolic linear layers are replaced with the tangent linear layers defined by Ganea et al. (2018); HAEGAN-$\beta$, where the concatenation in HAEGAN is replaced by $\beta$-concatenation (Shimizu et al., 2021); HAEGAN-T, where the concatenation is replaced by the Lorentz tangent concatenation discussed in §4.

**Metrics** We use the following metrics from You et al. (2018b) to evaluate the models: The Maximum Mean Discrepancy (MMD) of degree distribution (*Degree*), MMD of the orbit counts statistics distribution (*Orbit*); Average difference of orbit counts statistics (*Orbit*), Betweenness Centrality (*Betweenness*), Closeness Centrality (*Closeness*). All metrics are calculated between the test dataset and 100 samples generated from the models.

**Results** The results and runtime of all models are shown in Table 1. For all metrics, a smaller number implies a better result.

Our default choice of HAEGAN (with direct concatenation and fully hyperbolic linear layers) performs the best across all metrics except the MMD of orbit counts statistics distribution, in which it just marginally falls behind the $\beta$-concatenation. In particular, the Lorentz direct concatenation generally performs better and more efficiently than Lorentz tangent concatenation and $\beta$-concatenation. Also, the fully hyperbolic linear layer is superior than the tangent linear layer in both effectiveness and efficiency. Our results also show the advantage of the overall framework compared with either a single GAN or VAE. On one hand, the performance of HAEGAN is much better than HGAN. On the other hand, we note that the hyperbolic VAE-based methods suffer from numerical instability for this simple dataset even when using fully hyperbolic linear layers and direct concatenation, possibly because of the complicated ELBO loss. Finally, we remark that it is clear from the results that the hyperbolic models are better at generating trees than the Euclidean ones.

Table 1: Results of the tree generation experiments. "NaN" indicates NaN reported during training.

| | Concat | HNN | MMD | | Average Difference | | | Time |
| | | | Degree | Orbit | Orbit | Betweenness | Closeness | (s/step) |
|---|---|---|---|---|---|---|---|---|
| HGAN | Direct | Fully | 0.000566 | 0.000056 | 0.131509 | 0.027145 | 0.022921 | 1.5472 |
| HVAE-w | Direct | Fully | NaN | NaN | NaN | NaN | NaN | 1.6712 |
| HVAE-r | Direct | Fully | NaN | NaN | NaN | NaN | NaN | 1.7394 |
| GraphRNN | N/A | N/A | 0.002681 | 0.000106 | 0.143869 | 0.025565 | 0.022051 | 0.0933 |
| AEGAN | N/A | N/A | 0.001343 | 0.000050 | 0.140482 | 0.025666 | 0.021855 | 1.1874 |
| HAEGAN-H | Direct | Tangent | 0.000743 | 0.000010 | 0.138211 | 0.024512 | 0.022037 | 1.8513 |
| HAEGAN-$\beta$ | Beta | Fully | 0.000470 | **0.000001** | 0.129896 | 0.026102 | 0.022375 | 1.7529 |
| HAEGAN-T | Tangent | Fully | 0.000314 | 0.000052 | 0.131563 | 0.024171 | 0.021858 | 1.6385 |
| HAEGAN | Direct | Fully | **0.000156** | 0.000005 | **0.123286** | **0.023706** | **0.021740** | 1.3146 |

## 5.2 DE NOVO MOLECULAR GENERATION WITH HAEGAN

It is a crucial task in machine learning to learn structure of molecules, which has important application in discovery of drugs and proteins (Elton et al., 2019). Since molecules naturally show a graph structure, many recent works use graph neural networks to extract their information and accordingly train molecular generators (Simonovsky & Komodakis, 2018; De Cao & Kipf, 2018; Jin et al., 2018; 2019). In particular, Jin et al. (2018; 2019) proposed a bi-level representation of molecules where both a junction-tree skeleton and a molecular graph are used to represent the original molecular data. In this way, a molecule is represented in a hierarchical manner with a tree-structured scaffold. Given that hyperbolic spaces can well-embed such hierarchical and tree-structured data (Peng et al., 2021), we expect that HAEGAN can leverage the structural information. To validate its effectiveness, in this section, we test HAEGAN using molecular generative tasks, where the latent distribution is embedded in a hyperbolic manifold.

In our experiments, we design both a hyperbolic tree AE and a hyperbolic graph AE in our HAE-GAN to embed the structural information of the atoms in each molecule. Specifically, our model takes a molecular graph as the input, passes the original graph to the graph encoder and feeds the corresponding junction tree to the tree encoder, acquiring hyperbolic latent representations $z_G$ of the graph, as well as $z_T$ for the junction tree. Then, the junction tree decoder constructs a tree from $z_T$ autoregressively. Finally, the graph decoder recovers the molecular graph using the generated junction tree and $z_G$. Within the HAEGAN framework, this network contains fully hyperbolic layers and embedding layers, as well as the concatenation layers described in §4. We describe the detailed structure of each component in HAEGAN and include a figure illustration in Appendix E. The hyperbolic representation is supposed to better leverage the hierarchical structure from the junction-tree than the graph neural networks (Jin et al., 2018; 2019).

**Dataset**   We train and test our model on the MOSES benchmarking platform (Polykovskiy et al., 2020), which is refined from the ZINC dataset (Sterling & Irwin, 2015) and contains about 1.58M training, 176k test, and 176k scaffold test molecules. The molecules in the scaffold test set have different Bemis-Murcko scaffolds (Bemis & Murcko, 1996), which represent the core structures of compounds, than both the training and the test set. They are used to determine whether a model can generate novel scaffolds absent in the training set.

**Baselines**   We compare our model with the following baselines: non-neural models including the Hidden Markov Model (HMM), the N-Gram generative model (NGram) and the combinatorial generator; and neural methods including CharRNN (Segler et al., 2018), AAE (Kadurin et al., 2017a;b; Polykovskiy et al., 2018), VAE (Gómez-Bombarelli et al., 2018; Blaschke et al., 2018), JTVAE (Jin et al., 2018), LatentGAN (Prykhodko et al., 2019). The benchmark results are taken from (Polykovskiy et al., 2020)[1].

**Ablations**   On one hand, we consider an Euclidean counterpart of HAEGAN, named as AEGAN, to examine whether the hyperbolic setting indeed contributes. The architecture of AEGAN is the same as HAEGAN, except that the hyperbolic layers are replaced with Euclidean ones.

On the other hand, we also report the following alternative hyperbolic methods: HVAE-w and HVAE-r, where we use the same tree and graph AE but follow the ELBO loss function used by Mathieu et al. (2019) instead of having a GAN ("w" and "r" refer to using wrapped and Riemannian normal distributions, respectively); HGAN, where we train an end-to-end hyperbolic WGAN with the graph and tree decoder as the generator, and the graph and tree encoder as the critic; HAEGAN-H, HAEGAN-$\beta$ and HAEGAN-T as introduced in §5.1.

**Metrics**   We briefly describe how the models are evaluated. Detailed descriptions of the following metrics can be found in the MOSES benchmarking platform (Polykovskiy et al., 2020). We generate a set of 30,000 molecules, which we call the "generated set". On one hand, we report the *Validity*, *Unique(ness)* and *Novelty* scores, which are the percentage of valid, unique and novel molecules in the generated set, respectively. These are standard metrics widely used to represent the quality of the generation. On the other hand, we evaluate the following structure-related metrics by comparing the generated set with the test set and the scaffold set: Similarity to a Nearest Neighbor (*SNN*) and the Scaffold similarity (*Scaf*). SNN is the average Tanimoto similarly (Tanimoto, 1958) between generated molecule and its nearest neighbor in the reference set. Scaf is cosine distances between the scaffold frequency vectors (Bemis & Murcko, 1996) of the generated and reference sets. In particular, SNN compares the detailed structures while Scaf compares the skeleton structures. By considering them with both the test and the scaffold test sets, we measure both the structural similarity to training data and the capability of searching for novel structures.

**Results**   We describe the detailed settings and architectures of HAEGAN in Appendix F.5 and present some generated examples in Appendix G. We report in Table 2 the performance of HAEGAN and the baselines. For each metric described above, we take the mean and standard deviation from three independent samples. For all the metrics, a larger number implies a better result. We use bold font to highlight the best performing model in each criteria.

---

[1]We take the most updated results available from `https://github.com/molecularsets/moses`.

Table 2: Performance in Validity, Unique(ness), Novelty, SNN, and Scaf metrics. Reported (mean ± std) over three independent samples. HVAE-w, HVAE-r, HGAN, HAEGAN-H, HAEGAN-$\beta$, HAEGAN-T are not included in the table as they all produce "NaN", implying instability.

| Model | Validity (↑) | Unique (↑) | Novelty (↑) | SNN (↑) | | Scaf (↑) | |
|---|---|---|---|---|---|---|---|
| | | | | Test | TestSF | Test | TestSF |
| *Train* | *1* | *1* | *1* | *0.6419* | *0.5859* | *0.9907* | *0* |
| HMM | 0.076±0.032 | 0.567±0.142 | **0.999±0.001** | 0.388±0.011 | 0.380±0.011 | 0.207±0.048 | 0.049±0.018 |
| NGram | 0.238±0.003 | 0.922±0.002 | 0.969±0.001 | 0.521±0.001 | 0.499±0.001 | 0.530±0.016 | 0.098±0.014 |
| Combinatorial | **1.0±0.0** | 0.991±0.001 | 0.988±0.001 | 0.451±0.001 | 0.439±0.001 | 0.445±0.006 | 0.087±0.003 |
| CharRNN | 0.975±0.026 | 0.999±0.001 | 0.842±0.051 | 0.602±0.021 | 0.565±0.014 | 0.924±0.006 | 0.110±0.008 |
| AAE | 0.937±0.034 | 0.997±0.002 | 0.793±0.029 | 0.608±0.004 | 0.568±0.005 | 0.902±0.038 | 0.079±0.009 |
| VAE | 0.977±0.001 | 0.998±0.001 | 0.695±0.007 | 0.626±0.001 | 0.578±0.001 | **0.939±0.002** | 0.059±0.010 |
| JTVAE | **1.0±0.0** | 0.999±0.001 | 0.914±0.009 | 0.548±0.008 | 0.519±0.007 | 0.896±0.004 | 0.101±0.011 |
| LatentGAN | 0.897±0.003 | 0.997±0.001 | 0.950±0.001 | 0.537±0.001 | 0.513±0.001 | 0.887±0.001 | 0.107±0.010 |
| HAEGAN (Ours) | **1.0±0.0** | **1.0±0.0** | 0.905±0.006 | **0.631±0.004** | **0.593±0.002** | 0.874±0.002 | **0.113±0.007** |
| AEGAN | **1.0±0.0** | 0.968±0.001 | 0.995±0.009 | 0.459±0.006 | 0.452±0.006 | 0.203±0.004 | 0.058±0.008 |

First of all, HAEGAN achieves perfect validity and uniqueness scores, which implies the hyperbolic embedding adopted by HAEGAN does not break the rule of molecule structures and does not induce mode collapse. Moreover, our model significantly outperforms the baseline models in the SNN metric. This means that the molecules generated by our model have a closer similarity to the reference set. It implies that our model captures better the underlying structure of the molecules and our hyperbolic latent space is more suitable for embedding molecules than its Euclidean counterparts. Our model also achieves competitive performance in the Scaf metric when the reference set is the scaffold test set. This shows that our model is better in searching on the manifold of scaffolds and can generate examples with novel core structures.

Next, although AEGAN can also achieve very good performance in validity, uniqueness and novelty, we notice the big margin HAEGAN has over AEGAN in the structure-related metrics. This suggests that working with the hyperbolic space is necessary in our approach and the hyperbolic space better represents structural information.

Lastly, the alternative hyperbolic models all suffer from numerical instability and training reports NaN. This is not surprising since hyperbolic neural operations are known to easily make training unstable, especially in deep and complex networks. The result reveals the stronger numerical stability of HAEGAN, which highlights the importance of (1) the overall framework of HAEGAN (v.s. HVAE); (2) the fully hyperbolic layers (v.s. HAEGAN-H); (3) the Lorentz direct concatenation (v.s. HAEGAN-$\beta$, HAEGAN-T).

# 6 CONCLUSION AND FUTURE WORK

In this paper, we proposed HAEGAN, a hybrid AE-GAN framework and showed its capability of generating faithful hyperbolic examples. We showed that HAEGAN is able to generate both synthetic hyperbolic data and real molecular data. In particular, HAEGAN delivers state-of-the-art results for molecular data in structure-related metrics. It is not only the first hyperbolic GAN model that achieves such effectiveness, but also a deep hyperbolic model that does not suffer from numerical instability. We expect that HAEGAN can be applied to broader scenarios due to the flexibility in designing the hyperbolic AE and the possibility of building deep models.

Despite the promising results, we point out two possible limitations of the current model. First, not all complex modules are directly compatible with HAEGAN. Indeed, if the gated recurrent units (GRU) were used in our molecular generation task, the complex structure of GRU would cause unstable training and that is why a hyperbolic linear layer is used instead. Nevertheless, we expect defining more efficient hyperbolic operations that incorporate recurrent operations may alleviate the problem and leave it to future work. Second, although the hyperbolic operations in HAEGAN do not require going back and forth between the hyperbolic and the tangent spaces, we need to use exponential maps when sampling from the wrapped normal distribution. We will also work on more efficient ways of sampling from the hyperbolic Gaussian.

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

APPENDIX

The Appendix is organized as follows. In §A we describe the preliminaries on hyperbolic geometry. In §B we describe the details of hyperbolic GAN and demonstrate generating toy distributions. In §C we show detailed results from MNIST generation using HAEGAN. In §D we present more analysis for the Lorentz direct concatenation. In §E we provide a detailed description of the AE used in HAEGAN for molecular generation. In §F we carefully describe all the experimental details and neural network architectures that we did not cover in the main text for reproducing our work. In §G we illustrate a subset of examples of molecules generated by HAEGAN.

## A  PRELIMINARIES

### A.1  HYPERBOLIC GEOMETRY

We describe some fundamental concepts in hyperbolic geometry related to this work.

**The Lorentz Model**  The Lorentz model $\mathbb{L}_K^n = (\mathcal{L}, \mathfrak{g})$ of an $n$ dimensional hyperbolic space with constant negative curvature $K$ is an $n$-dimensional manifold $\mathcal{L}$ embedded in the $(n+1)$-dimensional Minkowski space, together with the Riemannian metric tensor $\mathfrak{g} = \mathrm{diag}([-1, \mathbf{1}_n^\top])$, where $\mathbf{1}_n$ denotes the $n$-dimensional vector whose entries are all 1's. Every point in $\mathbb{L}_K^n$ is represented by $\boldsymbol{x} = \begin{bmatrix} x_t \\ \boldsymbol{x}_s \end{bmatrix}, x_t > 0, \boldsymbol{x}_s \in \mathbb{R}^n$ and satisfies $\langle \boldsymbol{x}, \boldsymbol{x} \rangle_{\mathcal{L}} = 1/K$, where $\langle \cdot, \cdot \rangle_{\mathcal{L}}$ is the Lorentz inner product induced by $\mathfrak{g}$:

$$\langle \boldsymbol{x}, \boldsymbol{y} \rangle_{\mathcal{L}} := \boldsymbol{x}^\top \mathfrak{g} \boldsymbol{y} = -x_t y_t + \boldsymbol{x}_s^\top \boldsymbol{y}_s, \ \boldsymbol{x}, \boldsymbol{y} \in \mathbb{L}_K^n. \tag{7}$$

**Geodesics and Distances**  Geodesics are shortest paths in a manifold, which generalize the notion of "straight lines" in Euclidean geometry. In particular, the length of a geodesic in $\mathbb{L}_K^n$ (the "distance") between $\boldsymbol{x}, \boldsymbol{y} \in \mathbb{L}_K^n$ is given by

$$d_{\mathcal{L}}(\boldsymbol{x}, \boldsymbol{y}) = \frac{1}{\sqrt{-K}} \cosh^{-1}(K \langle \boldsymbol{x}, \boldsymbol{y} \rangle_{\mathcal{L}}). \tag{8}$$

**Tangent Space**  For each point $\boldsymbol{x} \in \mathbb{L}_K^n$, the tangent space at $\boldsymbol{x}$ is $\mathcal{T}_{\boldsymbol{x}} \mathbb{L}_K^n := \{ \boldsymbol{y} \in \mathbb{R}^{n+1} \mid \langle \boldsymbol{y}, \boldsymbol{x} \rangle_{\mathcal{L}} = 0 \}$. It is a first order approximation of the hyperbolic manifold around a point $\boldsymbol{x}$ and is a subspace of $\mathbb{R}^{n+1}$. We denote $\|\boldsymbol{v}\|_{\mathcal{L}} = \sqrt{\langle \boldsymbol{v}, \boldsymbol{v} \rangle_{\mathcal{L}}}$ as the norm of $\boldsymbol{v} \in \mathcal{T}_{\boldsymbol{x}} \mathbb{L}_K^n$.

**Exponential and Logarithmic Maps**  The exponential and logarithmic maps are maps between hyperbolic spaces and their tangent spaces. For $\boldsymbol{x}, \boldsymbol{y} \in \mathbb{L}_K^n$ and $\boldsymbol{v} \in \mathcal{T}_{\boldsymbol{x}} \mathbb{L}_K^n$, the exponential map $\exp_{\boldsymbol{x}}^K(\boldsymbol{v}) : \mathcal{T}_{\boldsymbol{x}} \mathbb{L}_K^n \to \mathbb{L}_K^n$ maps tangent vectors to hyperbolic spaces by assigning $\boldsymbol{v}$ to the point $\exp_{\boldsymbol{x}}^K(\boldsymbol{v}) := \gamma(1)$, where $\gamma$ is the geodesic satisfying $\gamma(0) = \boldsymbol{x}$ and $\gamma'(0) = \boldsymbol{v}$. Specifically,

$$\exp_{\boldsymbol{x}}^K(\boldsymbol{v}) = \cosh(\phi)\boldsymbol{x} + \sinh(\phi)\frac{\boldsymbol{v}}{\phi}, \phi = \sqrt{-K}\|\boldsymbol{v}\|_{\mathcal{L}}. \tag{9}$$

The logarithmic map $\log_{\boldsymbol{x}}^K(\boldsymbol{y}) : \mathbb{L}_K^n \to \mathcal{T}_{\boldsymbol{x}} \mathbb{L}_K^n$ is the inverse map that satisfies $\log_{\boldsymbol{x}}^K(\exp_{\boldsymbol{x}}^K(\boldsymbol{v})) = \boldsymbol{v}$. Specifically,

$$\log_{\boldsymbol{x}}^K(\boldsymbol{y}) = \frac{\cosh^{-1}(\psi)}{\sqrt{-K}} \frac{\boldsymbol{y} - \psi\boldsymbol{x}}{\|\boldsymbol{y} - \psi\boldsymbol{x}\|_{\mathcal{L}}}, \psi = K \langle \boldsymbol{x}, \boldsymbol{y} \rangle_{\mathcal{L}}. \tag{10}$$

**Parallel Transport**  For two points $\boldsymbol{x}, \boldsymbol{y} \in \mathbb{L}_K^n$, the parallel transport from $\boldsymbol{x}$ to $\boldsymbol{y}$ defines a map $\mathrm{PT}_{\boldsymbol{x} \to \boldsymbol{y}}^K$, which "transports" a vector from $\mathcal{T}_{\boldsymbol{x}} \mathbb{L}_K^n$ to $\mathcal{T}_{\boldsymbol{y}} \mathbb{L}_K^n$ along the geodesic from $\boldsymbol{x}$ to $\boldsymbol{y}$. Parallel transport preserves the metric, i.e. $\forall \boldsymbol{u}, \boldsymbol{v} \in \mathcal{T}_{\boldsymbol{x}} \mathbb{L}_K^n, \langle \mathrm{PT}_{\boldsymbol{x} \to \boldsymbol{y}}^K(\boldsymbol{v}), \mathrm{PT}_{\boldsymbol{x} \to \boldsymbol{y}}^K(\boldsymbol{u}) \rangle_{\mathcal{L}} = \langle \boldsymbol{v}, \boldsymbol{u} \rangle_{\mathcal{L}}$. In particular, the parallel transport in $\mathbb{L}_K^n$ is given by

$$\mathrm{PT}_{\boldsymbol{x} \to \boldsymbol{y}}^K(\boldsymbol{v}) = \frac{\langle \boldsymbol{y}, \boldsymbol{v} \rangle_{\mathcal{L}}}{-1/K - \langle \boldsymbol{x}, \boldsymbol{y} \rangle_{\mathcal{L}}}(\boldsymbol{x} + \boldsymbol{y}). \tag{11}$$

## A.2 More Hyperbolic Layers

We reviewed the Lorentz linear layer and the centroid distance layer in the main text. In this section, we review more hyperbolic layers.

The notion of the "centroid" of a set of points is important in formulating attention mechanism and feature aggregation. In the Lorentz model, with the squared Lorentzian distance defined as $d_{\mathcal{L}}^2(\boldsymbol{x}, \boldsymbol{y}) = 2/K - 2\langle \boldsymbol{x}, \boldsymbol{y} \rangle_{\mathcal{L}}$, $\boldsymbol{x}, \boldsymbol{y} \in \mathbb{L}_K^n$, the *hyperbolic centroid* (Law et al., 2019) is defined to be the minimizer that solves $\min_{\boldsymbol{\mu} \in \mathbb{L}_K^n} \sum_{i=1}^N \nu_i d_{\mathcal{L}}^2(\boldsymbol{x}_i, \boldsymbol{\mu})$ subject to $\boldsymbol{x}_i \in \mathbb{L}_K^n$, $\nu_i \geq 0$, $\sum_i \nu_i > 0$, $i = 1, \cdots, N$. A closed form of the centroid is given by

$$\boldsymbol{\mu} = \text{HCent}(\boldsymbol{X}, \boldsymbol{\nu}) = \frac{\sum_{i=1}^N \nu_i \boldsymbol{x}_i}{\sqrt{-K} \left| \| \sum_{i=1}^N \nu_i \boldsymbol{x}_i \|_{\mathcal{L}} \right|}, \tag{12}$$

where $\boldsymbol{X}$ is the matrix whose $i$-th row is $\boldsymbol{x}_i$. Extracting the hyperbolic centroid following hyperbolic linear layers produces a hyperbolic graph convolutional network (GCN) layer (Chen et al., 2021):

$$\boldsymbol{x}_v^{(l)} = \text{HGCN}(\boldsymbol{X}^{(l-1)})_v = \text{HCent}(\{\text{HLinear}_{d_{l-1}, d_l}(\boldsymbol{x}_u^{(l-1)}) \mid u \in N(v)\}, \boldsymbol{1}) \tag{13}$$

where $\boldsymbol{x}_v^{(l)}$ is the feature of node $v$ in layer $l$, $d_l$ denotes the dimensionality of layer $l$, and $N(v)$ is the set of neighbor points of node $v$.

## A.3 Embedding from Euclidean to Hyperbolic Spaces

It is possible that a dataset is originally represented as Euclidean, albeit having a hierarchical structure. In this case, the most obvious way of processing is to use the exponential or logarithmic maps so that we can represent data in the hyperbolic space. In order to map $\boldsymbol{t} \in \mathbb{R}^n$ to the hyperbolic space $\mathbb{L}_K^m$, Nagano et al. (2019) add a zero padding to the front of $\boldsymbol{t}$ to make it a vector in $\mathcal{T}_{\boldsymbol{o}} \mathbb{L}_K^m$, and then apply the exponential map. This *Euclidean to Hyperbolic* (E2H) operation was originally used for sampling, but can also be generally used to map from the Euclidean to the hyperbolic spaces. Specifically,

$$\boldsymbol{y} = \text{E2H}_m(\boldsymbol{t}) = \exp_{\boldsymbol{o}}^K \left( \left[ \begin{smallmatrix} 0 \\ \boldsymbol{t} \end{smallmatrix} \right] \right), \tag{14}$$

where $\boldsymbol{o} = \left[ \sqrt{-1/K}, 0, \ldots, 0 \right]^\top$ is the hyperbolic origin.

For better expressivity, especially when the input $\boldsymbol{x} \in \mathbb{R}^n$ is one-hot, one can first map the input to a hidden embedding $\boldsymbol{h} = \boldsymbol{W}\boldsymbol{x} \in \mathbb{R}^m$ with a trainable embedding matrix $\boldsymbol{W} \in \mathbb{R}^{n \times m}$. Then, it is mapped to hyperbolic space by the E2H operation defined as above. That is,

$$\boldsymbol{y} = \text{HEmbed}_{n,m}(\boldsymbol{x}) = \text{E2H}(\boldsymbol{W}\boldsymbol{x}). \tag{15}$$

This layer was previously used by Nagano et al. (2019) for word embedding.

We remark that besides using the E2H operation, other embedding methods (Nickel & Kiela, 2017; 2018; Sala et al., 2018; Sonthalia & Gilbert, 2020) exist. We use E2H in our experiments since it is simple to incorporate it in HAEGAN.

## A.4 More Related Works

**Machine Learning in Hyperbolic Spaces** A central topic in machine learning is to find methods and architectures that incorporate the geometric structure of data (Bronstein et al., 2021). Due to the data representation capacity of the hyperbolic space, many machine learning methods have been designed for hyperbolic data. Such methods include hyperbolic dimensionality reduction (Chami et al., 2021) and kernel hyperbolic methods (Fang et al., 2021). Besides these works, deep neural networks have also been proposed in the hyperpolic domain. One of the earliest such model was the Hyperbolic Neural Network (Ganea et al., 2018) which works with the Poincaré ball model of the hyperbolic space. This was recently refined in the Hyperbolic Neural Network ++ (Shimizu et al., 2021). Another popular choice is to use the Lorentz model of the hyperbolic space (Chen et al., 2021; Yang et al., 2022). Our model also uses Lorentz space for numerical stability.

**Hyperbolic Graph Neural Networks** Graph neural networks (GNNs) are successful models for learning representations of graph data. Recent studies (Boguná et al., 2010; Krioukov et al., 2010; Sala et al., 2018; Sonthalia & Gilbert, 2020) have found that hyperbolic spaces are suitable for tree-like graphs and a variety of hyperbolic GNNs (Chami et al., 2019; Liu et al., 2019; Bachmann et al., 2020; Dai et al., 2021a; Chen et al., 2021) have been proposed. In particular, Chami et al. (2019); Liu et al. (2019); Bachmann et al. (2020) all performed message passing, the fundamental operation in GNNs, in the tangent space of the hyperbolic space. On the other hand, Dai et al. (2021a); Chen et al. (2021) designed fully hyperbolic operations so that message passing can be done completely in the hyperbolic space. Some recent works address special GNNs. For instance, Sun et al. (2021) applied a hyperbolic time embedding to temporal GNN, while Zhang et al. (2021) designed a hyperbolic graph attention network. We also notice the recent survey on hyperbolic GNNs by Yang et al. (2022).

**Molecular Generation** State-of-the-art methods for molecular generation usually treat molecules as abstract graphs whose nodes represent atoms and edges represent chemical bonds. Early methods for molecular graph generations usually generate adjacency matrices via simple multilayer perceptrons (Simonovsky & Komodakis, 2018; De Cao & Kipf, 2018). Recently, Jin et al. (2018; 2019) proposed to treat a molecule as a multiresolutional representation, with a junction-tree scaffold, whose nodes represent valid molecular substructures. Other molecular graph generation methods include (You et al., 2018a; Shi* et al., 2020). Methods that work with SMILES (Simplified Molecular Input Line Entry System) notations instead of graphs include (Segler et al., 2018; Gómez-Bombarelli et al., 2018; Blaschke et al., 2018; Kadurin et al., 2017a;b; Polykovskiy et al., 2018; Prykhodko et al., 2019). Since the hyperbolic space is promising for tree-like structures, hyperbolic GNNs have also been recently used for molecular generation (Liu et al., 2019; Dai et al., 2021a).

# B HYPERBOLIC GENERATIVE ADVERSARIAL NETWORKS

## B.1 DETAILS OF THE HYPERBOLIC GAN

**Wrapped normal distribution** The wrapped normal distribution is a hyperbolic distribution whose density can be evaluated analytically and is differentiable with respect to the parameters (Nagano et al., 2019). Given $\boldsymbol{\mu} \in \mathbb{L}_K^n$ and $\boldsymbol{\Sigma} \in \mathbb{R}^{n \times n}$, to sample $\boldsymbol{z} \in \mathbb{L}_K^n$ from the wrapped normal distribution $\mathcal{G}(\boldsymbol{\mu}, \boldsymbol{\Sigma})$, we first sample a vector $\tilde{\boldsymbol{v}}$ from the Euclidean normal distribution $\mathcal{N}(\boldsymbol{0}, \boldsymbol{\Sigma})$, then identify $\tilde{\boldsymbol{v}}$ as an element $\boldsymbol{v} \in \mathcal{T}_{\boldsymbol{o}} \mathbb{L}_K^n$ so that $\boldsymbol{v} = \begin{bmatrix} 0 \\ \tilde{\boldsymbol{v}} \end{bmatrix}$. We parallel transport this $\boldsymbol{v}$ to $\boldsymbol{u} = \mathrm{PT}_{\boldsymbol{o} \to \boldsymbol{\mu}}^K(\boldsymbol{v})$ and then finally map $\boldsymbol{u}$ to $\boldsymbol{z} = \exp_{\boldsymbol{\mu}}(\boldsymbol{u}) \in \mathbb{L}_K^n$.

**Generator and critic** The generator pushes forward a wrapped normal distribution $\mathcal{G}(\boldsymbol{o}, \boldsymbol{I})$ to a hyperbolic distribution via a cascading of hyperbolic linear layers. The critic aims to distinguish between fake data generated from the generator and real data. It contains a cascading of hyperbolic linear layers, and a centroid distance layer whose output is a score in $\mathbb{R}$.

**Training** We adopt the framework of Wasserstein GAN (Arjovsky et al., 2017), which aims to minimize the Wasserstein-1 ($W_1$) distance between the distribution pushed forward by the generator and the data distribution. Since $d_{\mathcal{L}}$ is a valid metric, the $W_1$ distance between two hyperbolic distribution $\mathbb{P}_r, \mathbb{P}_g$ defined on the Lorentz space is

$$W_1(\mathbb{P}_r, \mathbb{P}_g) = \inf_{\gamma \in \Pi(\mathbb{P}_r, \mathbb{P}_g)} \mathbb{E}_{(\boldsymbol{x}, \boldsymbol{y}) \sim \gamma}[d_{\mathcal{L}}(\boldsymbol{x}, \boldsymbol{y})], \tag{16}$$

where $\Pi(\mathbb{P}_r, \mathbb{P}_g)$ is the set of all joint distributions whose marginals are $\mathbb{P}_r$ and $\mathbb{P}_g$, respectively. By Kantorovich-Rubinstein duality (Villani, 2009), we have the following more tractable form of $W_1$ distance

$$W_1(\mathbb{P}_r, \mathbb{P}_g) = \sup_{\|D\|_L \leq 1} \mathbb{E}_{\boldsymbol{x} \sim \mathbb{P}_r}[D(\boldsymbol{x})] - \mathbb{E}_{\boldsymbol{x} \sim \mathbb{P}_g}[D(\boldsymbol{x})], \tag{17}$$

where the supremum is over all 1-Lipschitz functions $D : \mathbb{L}_K^n \to \mathbb{R}$, represented by the critic. To enforce the 1-Lipschitz constraint, we adopt a penalty term on the gradient following Gulrajani et al. (2017). The loss function is thus

$$L_{\mathrm{WGAN}} = \mathbb{E}_{\tilde{\boldsymbol{x}} \sim \mathbb{P}_g}[D(\tilde{\boldsymbol{x}})] - \mathbb{E}_{\boldsymbol{x} \sim \mathbb{P}_r}[D(\boldsymbol{x})] + \lambda \mathbb{E}_{\hat{\boldsymbol{x}} \sim \mathbb{P}_{\hat{x}}}\left[ \left( \|\nabla D(\hat{\boldsymbol{x}})\|_{\mathcal{L}} - 1 \right)^2 \right], \tag{18}$$

where $\nabla D(\hat{\boldsymbol{x}})$ is the Riemannian gradient of $D(\boldsymbol{x})$ at $\hat{\boldsymbol{x}}$, $\mathbb{P}_g$ is the generator distribution and $\mathbb{P}_r$ is the data distribution, $\mathbb{P}_{\hat{\boldsymbol{x}}}$ samples uniformly along the geodesic between pairs of points sampled from $\mathbb{P}_g$ and $\mathbb{P}_r$. Next, we prove Proposition 3.1 to validate this sampling regime. The proof is obtained by carefully transferring the Euclidean case (Gulrajani et al., 2017) to the hyperbolic space.

## B.2 PROOF OF PROPOSITION 3.1

*Proof.* For the optimal solution $f^*$, we have

$$\mathbb{P}_{(\boldsymbol{x},\boldsymbol{y})\sim\pi}\left(f^*(\boldsymbol{y}) - f^*(\boldsymbol{x}) = d_{\mathcal{L}}(\boldsymbol{y},\boldsymbol{x})\right) = 1. \tag{19}$$

Let $\psi(t) = f^*(\boldsymbol{x}_t) - f^*(\boldsymbol{x})$, $0 \le t, t' \le 1$. Following Gulrajani et al. (2017), it is clear that $\psi$ is $d_{\mathcal{L}}(\boldsymbol{x},\boldsymbol{y})$-Lipschitz, and $f^*(\boldsymbol{x}_t) - f^*(\boldsymbol{x}) = \psi(t) = td_{\mathcal{L}}(\boldsymbol{x},\boldsymbol{y})$, $f^*(\boldsymbol{x}_t) = f^*(\boldsymbol{x}) + td_{\mathcal{L}}(\boldsymbol{x},\boldsymbol{y}) = f^*(\boldsymbol{x}) + t\|\boldsymbol{v}_t\|_{\mathcal{L}}$.

Let $\boldsymbol{u}_t = \frac{\boldsymbol{v}_t}{d_{\mathcal{L}}(\boldsymbol{x},\boldsymbol{y})} \in \mathcal{T}\mathbb{L}_K^n$ be the unit speed directional vector of the geodesic at point $\boldsymbol{x}_t$. Let $\alpha : [-1,1] \to \mathbb{L}_K^n$ be a differentiable curve with $\alpha(0) = \boldsymbol{x}_t$ and $\alpha'(0) = \boldsymbol{u}_t$. Note that $\gamma'(t) = d_{\mathcal{L}}(\boldsymbol{x},\boldsymbol{y})\alpha'(0)$. Therefore,

$$\lim_{h\to 0}\alpha(h) = \lim_{h\to 0}\gamma\left(t + \frac{h}{d_{\mathcal{L}}(\boldsymbol{x},\boldsymbol{y})}\right) = \lim_{h\to 0}\boldsymbol{x}_{t+\frac{h}{d_{\mathcal{L}}(\boldsymbol{x},\boldsymbol{y})}}. \tag{20}$$

The directional derivative can be thus calculated as

$$
\begin{aligned}
\nabla_{\boldsymbol{u}_t}f^*(\boldsymbol{x}_t) &= \left.\frac{d}{d\tau}f^*(\alpha(\tau))\right|_{\tau=0} = \lim_{h\to 0}\frac{f^*(\alpha(h)) - f^*(\alpha(0))}{h} \\
&= \lim_{h\to 0}\frac{f^*\left(\boldsymbol{x}_{t+\frac{h}{d_{\mathcal{L}}(\boldsymbol{x},\boldsymbol{y})}}\right) - f^*(\boldsymbol{x}_t)}{h} \\
&= \lim_{h\to 0}\frac{f^*(\boldsymbol{x}) + (t + \frac{h}{d_{\mathcal{L}}(\boldsymbol{x},\boldsymbol{y})})d_{\mathcal{L}}(\boldsymbol{x},\boldsymbol{y}) - f^*(\boldsymbol{x}) - td_{\mathcal{L}}(\boldsymbol{x},\boldsymbol{y})}{h} \\
&= \lim_{h\to 0}\frac{h}{h} = 1.
\end{aligned} \tag{21}
$$

Since $f^*$ is 1-Lipschitz, we have $\|\nabla f^*(\boldsymbol{x}_t)\|_{\mathcal{L}} \le 1$. This implies

$$
\begin{aligned}
1 &\ge \|\nabla f^*(\boldsymbol{x})\|_{\mathcal{L}}^2 \\
&= \langle\boldsymbol{u}_t, \nabla f^*(\boldsymbol{x}_t)\rangle_{\mathcal{L}}^2 + \|\nabla f^*(\boldsymbol{x}_t) - \langle\boldsymbol{u}_t, \nabla f^*(\boldsymbol{x}_t)\rangle\boldsymbol{u}_t\|_{\mathcal{L}}^2 \\
&= |\nabla_{\boldsymbol{u}_t}f^*(\boldsymbol{x}_t)|^2 + \|\nabla f^*(\boldsymbol{x}_t) - \boldsymbol{u}_t\nabla_{\boldsymbol{u}_t}f^*(\boldsymbol{x}_t)\|_{\mathcal{L}}^2 \\
&= 1 + \|\nabla f^*(\boldsymbol{x}_t) - \boldsymbol{u}_t\|_{\mathcal{L}}^2 \ge 1.
\end{aligned} \tag{22}
$$

Therefore, we have $1 = 1 + \|\nabla f^*(\boldsymbol{x}_t) - \boldsymbol{u}_t\|_{\mathcal{L}}^2$, $\nabla f^*(\boldsymbol{x}_t) = \boldsymbol{u}_t$. This yields $\nabla f^*(\boldsymbol{x}_t) = \frac{\boldsymbol{v}_t}{d_{\mathcal{L}}(\boldsymbol{x},\boldsymbol{y})}$.
$\square$

## B.3 TOY DISTRIBUTIONS

We use a set of challenging toy 2D distributions explored by Rozen et al. (2021) to test the effectiveness of the hyperbolic GAN. We create the dataset in the same way using their code[2]. For our experiment, the training data are prepared in the following manner. We first sample 5,000 points from the toy 2D distributions and scale the coordinates to $[-1, 1]$. Then, we use the E2H operation (14) to map the points to the hyperbolic space. These points are treated as the input data of the hyperbolic GAN. Next, we use the hyperbolic GAN to learn the hyperbolic toy distributions. The generator and the critic both contain 3 layers of hyperbolic linear layers and 64 hidden dimensions at

---

[2]https://github.com/noamroze/moser_flow

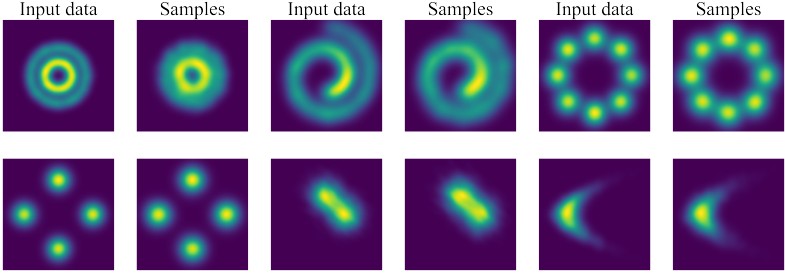

Figure 4: Input data and generated samples from the hyperbolic GAN. The hyperbolic data points are transformed to the tangent space of the origin by the logarithmic map.

each layer. The input dimension for the generator is 128. We present the training details in Appendix F.3.

After we train the hyperbolic GAN, we sample from it and compare with the input data. Note that the input data and the generated samples are both in the hyperbolic space. To illustrate them, we map both the input data and the generated samples to the tangent space of the origin by applying the logarithmic map. We present the mapped input data and generated samples in Figure 4. Clearly, the hyperbolic GAN can faithfully represent the challenging toy distributions in the hyperbolic space.

## C   MORE RESULTS FROM MNIST GENERATION WITH HAEGAN

In the HAEGAN for generating MNIST, the encoder of the AE consists of three convolutional layers, followed by an E2H layer and three hyperbolic linear layers, while the decoder consists of three hyperbolic linear layers, a logarithmic map to the Euclidean space, and three deconvolutional layers. We describe the training procedures as follows. Firstly, we normalize the MNIST dataset and train the AE by minimizing the reconstruction loss. Secondly, we use the encoder to embed the MNIST in hyperbolic space and train the hyperbolic GAN with the hyperbolic embedding. Finally, we sample a hyperbolic embedding using the generator and use it to produce an image by applying the decoder. We describe the detailed architecture and settings in Appendix F.4.

The training curves of the hyperbolic GAN of HAEGAN in the MNIST generation task are shown in Figure 5, which we compare with an Euclidean Wasserstein GAN. The critic loss includes the gradient penalty term. We observe that the trend of loss in the hyperbolic GAN is similar to the Euclidean one (both the generator and critic) and no instability from the hyperbolic model.

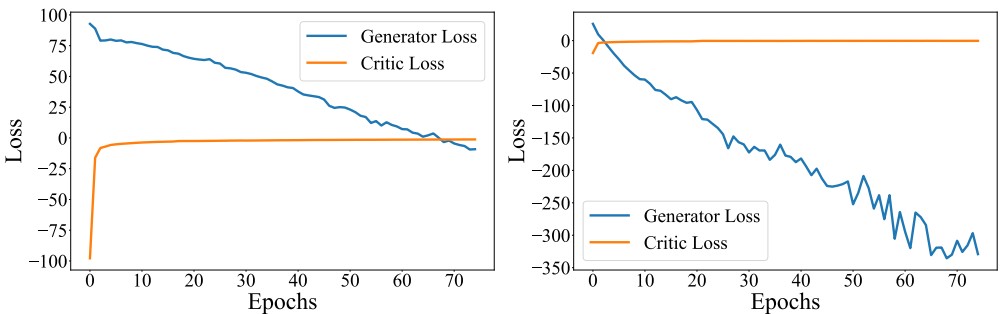

Figure 5: The training loss of generator and critic in the MNIST generation task. Left: hyperbolic Wasserstein GAN. Right: Euclidean Wasserstein GAN.

In Table 3 and Table 4, we report the quantitative results for the MNIST generation task. First, we compare the negative log-likelihood (NLL) results between our method and hyperbolic VAEs (Mathieu et al., 2019; Nagano et al., 2019; Bose et al., 2020). The results are directly taken from

the respective papers, which were produced from different numbers of samples: in Nagano et al. (2019), the NLL is calculated with 500 samples, while Mathieu et al. (2019) used 3,000 samples. We generate 5,000 samples and compare the NLL with them. Then, we also calculate the FID and compare it with HGAN (Lazcano et al., 2021). Our NLL results are comparable with the hyperbolic VAEs while FID is slightly better than HGAN.

Table 3: Quantitative comparison between HAEGAN and other methods in the MNIST generation task. Log Likelihood (±std) for different embedding dimensions is reported. We use 5000 samples to estimate the log-likelihood. Results for (Mathieu et al., 2019; Nagano et al., 2019; Bose et al., 2020) are taken directly from the respective paper. ⋆ indicates numerically unstable settings.

| Model | Dimensionality | | | |
|---|---|---|---|---|
| | 2 | 5 | 10 | 20 |
| $\mathcal{N}$-VAE (Mathieu et al., 2019) | -144.5±0.4 | -114.7±0.1 | -100.2±0.1 | -97.6±0.1 |
| $\mathcal{P}$-VAE (Wrapped) (Mathieu et al., 2019) | -143.8±0.6 | -114.7±0.1 | -100.0±0.1 | -97.1±0.1 |
| $\mathcal{P}$-VAE (Riemannian) (Mathieu et al., 2019) | -142.5±0.4 | -114.1±0.2 | -99.7±0.1 | -97.0±0.1 |
| Vanilla VAE (Nagano et al., 2019) | -140.45±0.47 | -105.78±0.51 | -86.25±0.52 | -77.89±0.36 |
| Hyperbolic VAE (Nagano et al., 2019) | -138.61±0.45 | -105.38±0.61 | -86.40±0.28 | -79.23±0.20 |
| AEGAN (Ours) | -141.13±0.49 | -111.41±0.48 | -97.83±0.37 | -90.18±0.41 |
| HAEGAN (Ours) | -140.24±0.55 | -111.29±0.59 | -98.15±0.41 | -91.37±0.39 |
| | 2 | 4 | 6 | |
| $\mathcal{N}$-VAE (Bose et al., 2020) | -139.5±1.0 | -115.6±0.2 | -100.0±0.02 | |
| $\mathbb{H}$-VAE (Bose et al., 2020) | ⋆ | -113.7±0.9 | -99.8±0.2 | |
| $\mathcal{N}$C (Bose et al., 2020) | -139.2±0.4 | -115.2±0.6 | -98.70.3 | |
| $\mathcal{T}$C (Bose et al., 2020) | ⋆ | -112.5±0.2 | -99.3±0.2 | |
| $\mathcal{W}\mathbb{H}$C (Bose et al., 2020) | -136.5±2.1 | -112.8±0.5 | -99.4±0.2 | |

Table 4: Quantitative comparison between HAEGAN and HGAN (Lazcano et al., 2021) in the MNIST generation task. Fréchet inception distance (±std) is reported. Results for (Lazcano et al., 2021) are taken directly from the paper.

| Model | FID |
|---|---|
| HGAN (Lazcano et al., 2021) | 54.95 |
| HWGAN (Lazcano et al., 2021) | 12.50 |
| HCGAN (Lazcano et al., 2021) | 12.43 |
| HAEGAN (Ours) | 8.05±0.37 |

## D  MORE ANALYSIS OF CONCATENATION

### D.1  PROOF OF THEOREM 4.1

*Proof.* 1. First, consider the case where $y*$ is one of the spatial components of $\boldsymbol{y}$. According to (5), $y^*$ is a copy of an entry in $\boldsymbol{x}_{i_s}$ for some $i \in \{1, \cdots, N\}$. Therefore, if $i \neq j$, $\partial y^* / \partial \boldsymbol{x}_{j_s}$ is a zero vector; if $i = j$, $\partial y^* / \partial \boldsymbol{x}_{j_s}$ a one-hot vector. In both cases, $\|\partial y^* / \partial \boldsymbol{x}_{j_s}\| \leq 1$.

Next, consider the case where $y^* = y_t$ is the time component of $\boldsymbol{y}$. According to (5), $\partial y^* / \partial \boldsymbol{x}_{j_s} = \dfrac{\boldsymbol{x}_{j_s}}{\sqrt{\sum_{i=1}^N \|\boldsymbol{x}_{i_s}\|^2 - \frac{1}{K}}}$ and thus $\left\| \dfrac{\partial y^*}{\partial \boldsymbol{x}_{j_s}} \right\| = \dfrac{\|\boldsymbol{x}_{j_s}\|}{\sqrt{\sum_{i=1}^N \|\boldsymbol{x}_{i_s}\|^2 - \frac{1}{K}}} \leq 1$ since the curvature $K < 0$.

We conclude that $\|\partial y^* / \partial \boldsymbol{x}_{j_s}\| \leq 1$ for any entry $y^*$ in $\boldsymbol{y}$.

2. Write $\boldsymbol{z} = \left[z_t, \boldsymbol{z}_{1_s}^\top, \cdots, \boldsymbol{z}_{N_s}^\top\right]^\top$. According to the definition of the Lorentz tangent concatenation as well as formulas (9) and (10), for $l \in \{1, \cdots, N\}$, the $p$-th entry of the vector $\boldsymbol{z}_{l_s}$ can be written

in the following concrete form:

$$z_{l_s p} = \frac{\sinh(\Delta)}{\Delta} \frac{\cosh^{-1}\left(-K\sqrt{\|\boldsymbol{x}_{l_s}\|^2 - \frac{1}{K}}\right)}{\sqrt{-K}} \frac{x_{l_s p}}{\|\boldsymbol{x}_{l_s}\|}$$

$$= \frac{\sinh(\Delta)}{\Delta} \frac{\sinh^{-1}\left(\sqrt{K^2 \|\boldsymbol{x}_{l_s}\|^2 - K - 1}\right)}{\sqrt{-K}} \frac{x_{l_s p}}{\|\boldsymbol{x}_{l_s}\|}, \tag{23}$$

where

$$\Delta = \sqrt{-K} \left( \sum_{i=1}^{N} \left( \sinh^{-1}\left( \sqrt{K^2 \|\boldsymbol{x}_{i_s}\|^2 - K - 1} \right) \right)^2 \right)^{1/2}. \tag{24}$$

Let $j \neq l$. Differentiating $z_{l_s p}$ with respect to $x_{j_s k}$, the $k$-th entry of $\boldsymbol{x}_{j_s}$, yields

$$\frac{\partial z_{l_s p}}{\partial x_{j_s k}} = C_l \left( \frac{\cosh(\Delta)}{\Delta^2} - \frac{\sinh(\Delta)}{\Delta^3} \right) \cdot \sinh^{-1}\left( \sqrt{K^2 \|\boldsymbol{x}_{j_s}\|^2 - K - 1} \right) \cdot$$

$$\frac{1}{\sqrt{\|\boldsymbol{x}_{j_s}\|^2 - \frac{1}{K}}} \frac{x_{j_s k}}{\sqrt{\|\boldsymbol{x}_{j_s}\|^2 - \frac{1+K}{K^2}}}, \tag{25}$$

where

$$C_l = \sinh^{-1}\left( \sqrt{K^2 \|\boldsymbol{x}_{l_s}\|^2 - K - 1} \right) \frac{x_{l_s p}}{\|\boldsymbol{x}_{l_s}\|} \tag{26}$$

does not depend on $\boldsymbol{x}_{j_s}$.

Arbitrarily take fixed $\boldsymbol{x}_{i_s}$ for all $i \neq l$ with particularly $x_{j_s k} > 0$. Also arbitrarily take fixed $x_{l_s q}$ for $q \neq p$. We claim that $\dfrac{\partial z_{l_s p}}{\partial x_{j_s k}} \to \infty$ as $x_{l_s q} \to \infty$.

To prove the claim, first note that

$$\frac{\partial C_l}{\partial x_{l_s p}} = \frac{x_{l_s p}^2}{\|\boldsymbol{x}_{l_s}\|^2 \sqrt{\|\boldsymbol{x}_{j_s}\|^2 - \frac{1}{K}} \sqrt{\|\boldsymbol{x}_{j_s}\|^2 - \frac{1+K}{K^2}}} +$$

$$\left( 1 - \frac{x_{l_s p}^2}{\|\boldsymbol{x}_{l_s}^2\|} \right) \frac{\sinh^{-1}\left( \sqrt{K^2 \|\boldsymbol{x}_{l_s}\|^2 - K - 1} \right)}{\|\boldsymbol{x}_{l_s}\|}. \tag{27}$$

Since $x_{l_s p}^2 \leq \|\boldsymbol{x}_{l_s}\|^2$ and $\frac{\sinh^{-1}\left(\sqrt{K^2 \|\boldsymbol{x}_{l_s}\|^2 - K - 1}\right)}{\|\boldsymbol{x}_{l_s}\|} > 0$, the second term in (27) is positive. Consequently, $\partial C_l / \partial x_{l_s p} > 0$ and thus $C_l$ is a positive term that increases with $x_{l_s p}$. Moreover, $\sinh^{-1}\left( \sqrt{K^2 \|\boldsymbol{x}_{j_s}\|^2 - K - 1} \right) \dfrac{1}{\sqrt{\|\boldsymbol{x}_{j_s}\|^2 - 1/K}} \dfrac{x_{j_s k}}{\sqrt{\|\boldsymbol{x}_{j_s}\|^2 - \frac{1+K}{K^2}}}$ in (25) is a fixed positive term that does not depend on $x_{l_s p}$. Hence, we only need to show that $\dfrac{\cosh(\Delta)}{\Delta^2} - \dfrac{\sinh(\Delta)}{\Delta^3} \to \infty$ as $x_{l_s p} \to \infty$. Since $\Delta \to \infty$ as $x_{l_s p} \to \infty$, this reduces to proving $f(t) = \dfrac{\cosh(t)}{t^2} - \dfrac{\sinh(t)}{t^3} \to \infty$ as $t \to \infty$, which is an immediate result once we write explicitly that

$$f(t) = \frac{1}{2} \left( \frac{e^{-t}}{t^3} + \frac{e^{-t}}{t^2} + \frac{(t-1)e^t}{t^3} \right). \tag{28}$$

We conclude that for any $M > 0$, there exist $\{\boldsymbol{x}_i\}_{i=1}^{N}$ and an entry $z^*$ of $\boldsymbol{z}$, i.e. $z_{l_s p}$ above, for which all entries of $\partial z^* / \partial \boldsymbol{x}_{j_s}$ are no less than $M$. Thus it also holds that $\|\partial z^* / \partial \boldsymbol{x}_{j_s}|_{\boldsymbol{x}_1, \cdots, \boldsymbol{x}_N}\| \geq M$. $\quad\square$

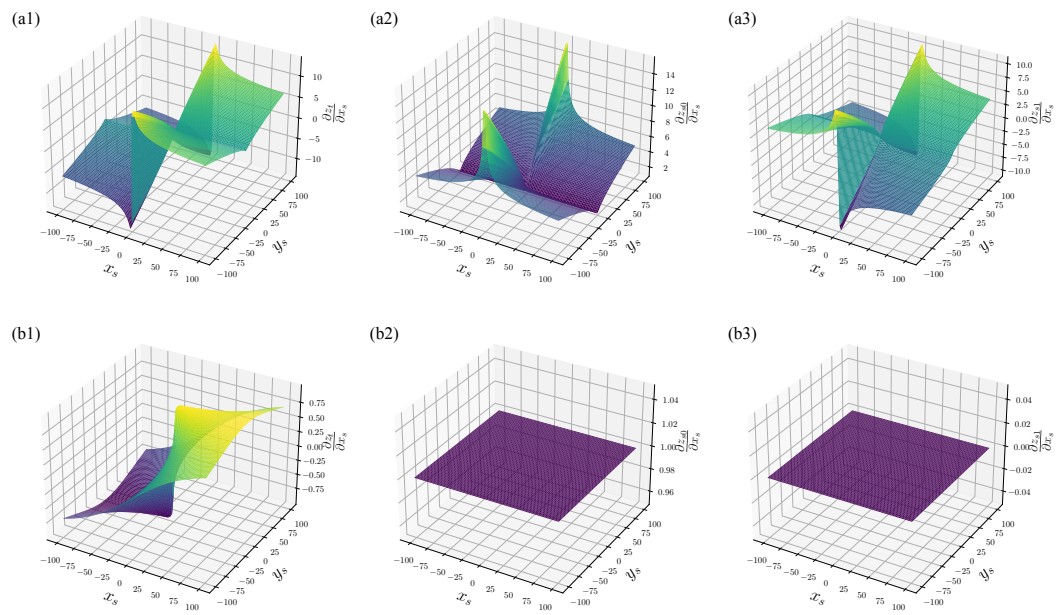

Figure 6: Illustration of the gradients of both concatenation methods: (a) Lorentz tangent concatenation (b) Lorentz direct concatenation. (a1) & (b1) $\dfrac{\partial z_t}{\partial x_s}$; (a2) & (b2) $\dfrac{\partial z_{s0}}{\partial x_s}$; (a3) & (b3) $\dfrac{\partial z_{s1}}{\partial x_s}$.

### D.2 GRADIENT OF CONCATENATION

In this section, we numerically validate Theorem 4.1 by showing the gradients of the Lorentz tangent concatenation and the Lorentz direct concatenation under the following simple setting. We concatenate two 1-dimensional Lorentz vectors to obtain a 2-dimensional Lorentz vector.

Denote $\boldsymbol{x} = \left[\sqrt{x_s^2+1}, x_s\right]^\top \in \mathbb{L}_{-1}^1$ and $\boldsymbol{y} = \left[\sqrt{y_s^2+1}, y_s\right]^\top \in \mathbb{L}_{-1}^1$ to be the two input vectors under the Lorentz model with $K = -1$. For both the Lorentz tangent concatenation and the Lorentz direct concatenation, let $\boldsymbol{z} = [z_t, z_{s0}, z_{s1}]^\top \in \mathbb{L}_{-1}^2$ denote the concatenated vector of $\boldsymbol{x}$ and $\boldsymbol{y}$. Numerically, we consider the range of $x_s, y_s \in [-100, 100]$ and calculate the gradients of each entry of $\boldsymbol{z}$ with respect to $x_s$, that is, $\dfrac{\partial z_t}{\partial x_s}\bigg|_{x_s, y_s \in [-100,100]}$, $\dfrac{\partial z_{s0}}{\partial x_s}\bigg|_{x_s, y_s \in [-100,100]}$ and $\dfrac{\partial z_{s1}}{\partial x_s}\bigg|_{x_s, y_s \in [-100,100]}$.

We plot them in Figure 6. We remark that due to symmetry, the graphs are the same for $\dfrac{\partial}{\partial y_s}$.

From Figure 6, we observe unbounded gradients in each component of the gradient of the Lorentz tangent concatenation. On the other hand, all the components of the gradient of the Lorentz direct concatenation has an absolute value bounded by 1. The results of this numerical experiment have validated the conclusion in Theorem 4.1.

### D.3 ANALYSIS OF EFFECT ON HYPERBOLIC DISTANCES

In this section, we perform additional analysis of Lorentz direct concatenation and Lorentz tangent concatenation, particularly their effect on hyperbolic distances.

First, we study the hyperbolic distances to the hyperbolic origin for both concatenation methods. Suppose we have $\boldsymbol{x} \in \mathbb{L}_K^n$ and $\boldsymbol{y} \in \mathbb{L}_K^m$. Let $\boldsymbol{z} = \mathrm{HCat}(\boldsymbol{x}, \boldsymbol{y}) \in \mathbb{L}_K^{n+m-1}$ and $\boldsymbol{z}' = \mathrm{HTCat}(\boldsymbol{x}, \boldsymbol{y}) \in \mathbb{L}_K^{n+m-1}$ be their hyperbolic direct concatenation and hyperbolic tangent concatenation, respectively. We compare the difference between $d_{\mathcal{L}}(\boldsymbol{z}, \boldsymbol{o})$ and $d_{\mathcal{L}}(\boldsymbol{z}', \boldsymbol{o})$ as follows. Note that the distance between an arbitrary point $\boldsymbol{x} \in \mathbb{L}_K^n$ and the origin only depend on the time

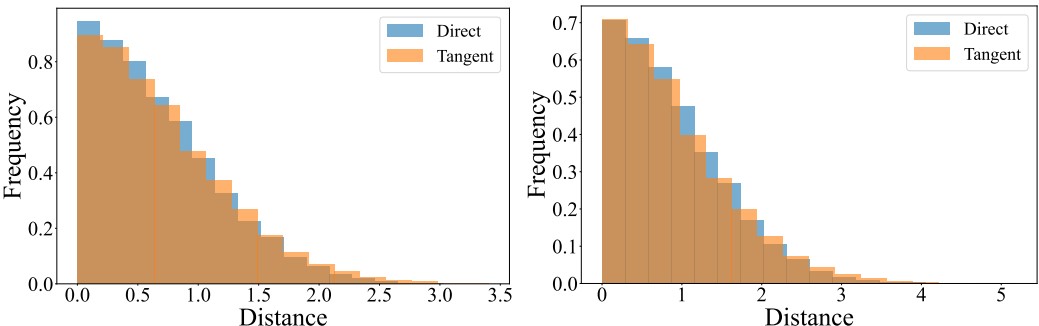

Figure 7: Difference between concatenated distances and original distances with $n = 3$. Left: spatial normal. Right: wrapped normal.

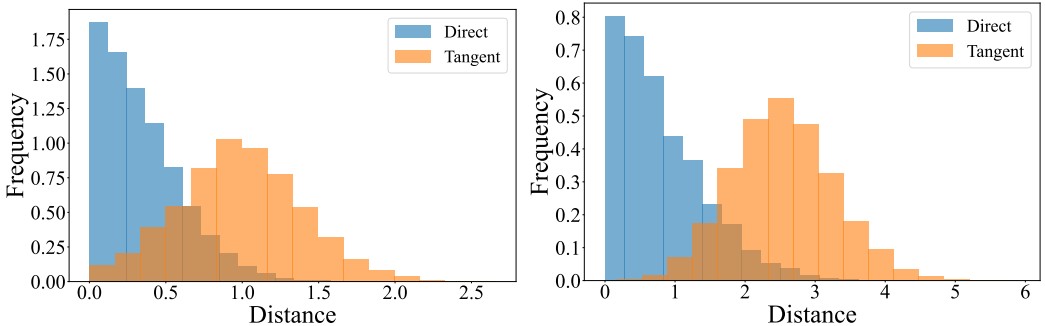

Figure 8: Difference between concatenated distances and original distances with $n = 16$. Left: spatial normal. Right: wrapped normal.

component:

$$d_{\mathcal{L}}(\boldsymbol{x}, \boldsymbol{o}) = \frac{1}{\sqrt{-K}} \cosh^{-1}(K\langle \boldsymbol{x}, \boldsymbol{o}\rangle_{\mathcal{L}}) = \frac{1}{\sqrt{-K}} \cosh^{-1}(-Kx_t). \tag{29}$$

Hence, the distance information is completely contained the time component. After the concatenation, the time component is $\sqrt{x_t^2 + y_t^2 + 1/K}$. Consequently, for Lorentz direct concatenation, the distance is

$$d_{\mathcal{L}}(\boldsymbol{z}, \boldsymbol{o}) = \frac{1}{\sqrt{-K}} \cosh^{-1}\left(-K\sqrt{x_t^2 + y_t^2 + 1/K}\right). \tag{30}$$

For Lorentz tangent concatenation, since both the logarithmic and exponential maps reserve the distances, one has

$$\begin{aligned} d_{\mathcal{L}}(\boldsymbol{z}', \boldsymbol{o}) &= \sqrt{d_{\mathcal{L}}(\boldsymbol{x}, \boldsymbol{o})^2 + d_{\mathcal{L}}(\boldsymbol{y}, \boldsymbol{o})^2} \\ &= \sqrt{\frac{1}{-K}\left(\cosh^{-2}(-Kx_t) + \cosh^{-2}(-Ky_t)\right)}. \end{aligned} \tag{31}$$

Although the hyperbolic distance $d_{\mathcal{L}}(\boldsymbol{z}, o)$ is not the squared sum of $d_{\mathcal{L}}(\boldsymbol{x}, \boldsymbol{o})$ and $d_{\mathcal{L}}(\boldsymbol{y}, \boldsymbol{o})$, $d_{\mathcal{L}}(\boldsymbol{z}, \boldsymbol{o})$ is larger than each of $d_{\mathcal{L}}(\boldsymbol{x}, \boldsymbol{o})$ and $d_{\mathcal{L}}(\boldsymbol{y}, \boldsymbol{o})$. On the other hand, after concatenation, $d_{\mathcal{L}}^2(\boldsymbol{z}', \boldsymbol{o}) = d_{\mathcal{L}}^2(\boldsymbol{x}, \boldsymbol{o}) + d_{\mathcal{L}}^2(\boldsymbol{y}, \boldsymbol{o})$. This relation agrees with the Euclidean concatenation. However, norm-preservation is not why concatenation works in the Euclidean domain. Therefore, we don't consider this as an advantage of the Lorentz tangent concatenation. The Lorentz direct concatenation is more efficient and stable, and no information is lost during concatenation. Therefore, it is still preferred as a neural layer.

More importantly, we study how concatenation changes the relative distances, which is closely related to stability. Specifically, we perform the following experiments. Given $\boldsymbol{x}, \boldsymbol{y}, \boldsymbol{c} \in \mathbb{L}_K^n$,

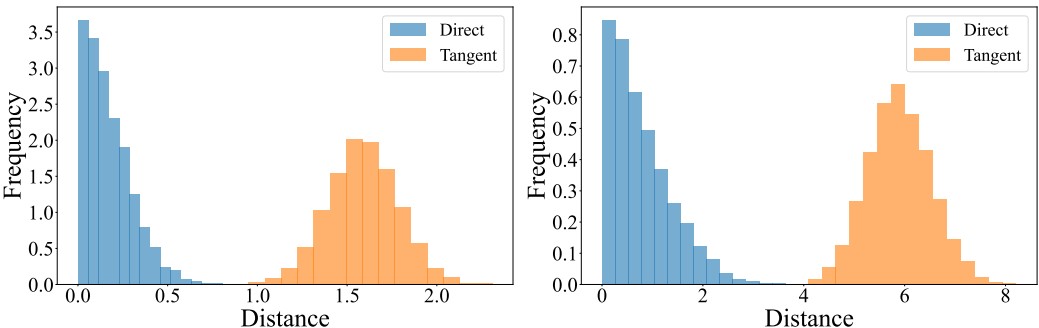

Figure 9: Difference between concatenated distances and original distances with $n = 64$. Left: spatial normal. Right: wrapped normal.

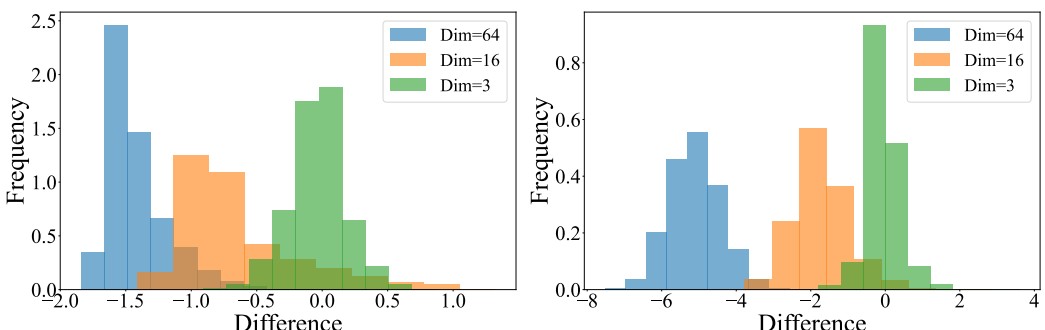

Figure 10: $|d_{\mathcal{L}}(\boldsymbol{x}_c, \boldsymbol{y}_c) - d_{\mathcal{L}}(\boldsymbol{x}, \boldsymbol{y})| - |d_{\mathcal{L}}(\boldsymbol{x}'_c, \boldsymbol{y}'_c) - d_{\mathcal{L}}(\boldsymbol{x}, \boldsymbol{y})|$. Left: spatial normal. Right: wrapped normal.

let $\boldsymbol{x}_c = \mathrm{HCat}(\boldsymbol{x}, \boldsymbol{c})$ be the direct-concatenated version of $\boldsymbol{x}$ and $\boldsymbol{c}$, while $\boldsymbol{y}_c = \mathrm{HCat}(\boldsymbol{y}, \boldsymbol{c})$ be the direct-concatenated version of $\boldsymbol{y}$ and $\boldsymbol{c}$. Similarly we denote $\boldsymbol{x}'_c = \mathrm{HTCat}(\boldsymbol{x}, \boldsymbol{c})$ and $\boldsymbol{y}'_c = \mathrm{HTCat}(\boldsymbol{y}, \boldsymbol{c})$. Since the same vector $\boldsymbol{c}$ is attached to $\boldsymbol{x}$ and $\boldsymbol{y}$, we naturally hope $d_{\mathcal{L}}(\boldsymbol{x}_c, \boldsymbol{y}_c)$ and $d_{\mathcal{L}}(\boldsymbol{x}'_c, \boldsymbol{y}'_c)$ do not deviate much from $d_{\mathcal{L}}(\boldsymbol{x}, \boldsymbol{y})$.

We describe our experiments as follows. Take $K = -1$. We randomly sample three points independently from $\mathbb{L}_K^n$ as $\boldsymbol{x}$, $\boldsymbol{y}$ and $\boldsymbol{c}$ respectively. We have two scenarios for sampling the points: (1) "spatial normal": the points are sampled so that their spatial components follow the standard normal distribution; (2) "wrapped normal": the points are sampled from the wrapped normal distribution with unit variance. In each scenario, for $n \in \{3, 16, 64\}$, we do the experiments for 10,000 times. We report the distances $|d_{\mathcal{L}}(\boldsymbol{x}_c, \boldsymbol{y}_c) - d_{\mathcal{L}}(\boldsymbol{x}, \boldsymbol{y})|$ and $|d_{\mathcal{L}}(\boldsymbol{x}'_c, \boldsymbol{y}'_c) - d_{\mathcal{L}}(\boldsymbol{x}, \boldsymbol{y})|$ in Figures 7–9, as well as their differences in Figure 10.

Our experiments clearly show that, especially for large dimensions, the distance between $d_{\mathcal{L}}(\boldsymbol{x}_c, \boldsymbol{y}_c)$ and $d_{\mathcal{L}}(\boldsymbol{x}, \boldsymbol{y})$ is smaller than the distance between $d_{\mathcal{L}}(\boldsymbol{x}'_c, \boldsymbol{y}'_c)$ and $d_{\mathcal{L}}(\boldsymbol{x}, \boldsymbol{y})$. In particular, in many cases, $|d_{\mathcal{L}}(\boldsymbol{x}_c, \boldsymbol{y}_c) - d_{\mathcal{L}}(\boldsymbol{x}, \boldsymbol{y})|$ is around zero. On the other hand, $|d_{\mathcal{L}}(\boldsymbol{x}'_c, \boldsymbol{y}'_c) - d_{\mathcal{L}}(\boldsymbol{x}, \boldsymbol{y})|$ tend to be large when $n = 16, 64$, especially when samples follow the wrapped normal distribution. From this result, the Lorentz Direct Concatenation should be preferred to the Lorentz Tangent Concatenation. In particular, the significant expansion of distance when concatenating with the same vector, in the case $n = 64$, may be one cause of numerical instability (note that we use the same dimensionality for the experiment in §4.2 and in the molecular generation task).

# E DETAILS OF AE IN MOLECULAR GENERATION

In this section, we carefully describe the encoders and decoders, which compose the AE used in the HAEGAN for molecular generation. The basic structure of the AE in the molecular generation task is illustrated in Figure 11.

**Notation** We denote a molecular graph as $G = (V_G, E_G)$, where $V_G$ is the set of nodes (atoms) and $E_G$ is the set of edges (bonds). Each node (atom) $v \in V_G$ has a node feature $\boldsymbol{x}_v$ describing its atom type and properties. The molecular graph is decomposed into a junction tree $T = (V_T, E_T)$ where $V_T$ is the set of atom clusters. We use $u, v, w$ to represent graph nodes and $i, j, k$ to represent tree nodes, respectively. The dimensions of the node features of the graph $\boldsymbol{x}_v$ and the tree $\boldsymbol{x}_i$ are denoted by $d_{G_0}$ and $d_{T_0}$, respectively. The hidden dimensions of graph and tree embeddings are $d_G$, $d_T$, respectively.

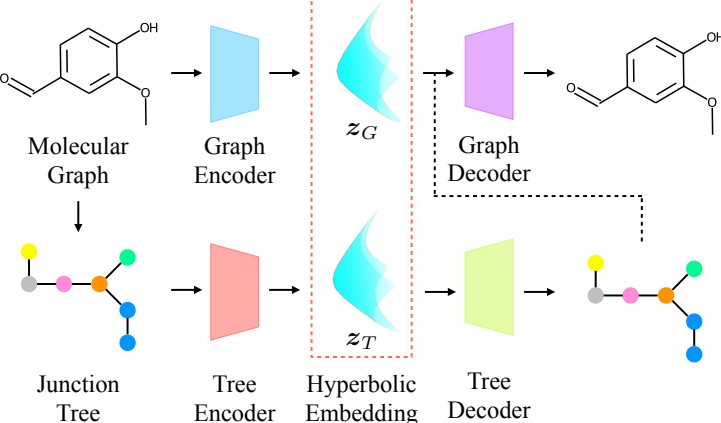

Figure 11: Illustration of the autoencoder used in the HAEGAN for molecular generation. The input molecular graph is firstly coarsened into the junction tree. Then both of them are encoded using graph and tree encoders to their respective hyperbolic embeddings $\boldsymbol{z}_T$ and $\boldsymbol{z}_G$. To reconstruct the molecule, we first decode the junction tree from $\boldsymbol{z}_T$, and then reconstruct the molecular graph using the junction tree and $\boldsymbol{z}_G$.

## E.1 GRAPH AND TREE ENCODER

The graph encoder for the molecular graph $G$ contains the following layers. First, each node feature $\boldsymbol{x}_v$ is mapped to the hyperbolic space via

$$\boldsymbol{x}_v^{(0)} = \text{E2H}_{d_{G_0}}(\boldsymbol{x}_v). \tag{32}$$

Next, the hyperbolic feature is passed to a hyperbolic GCN with $l_G$ layers

$$\boldsymbol{x}^{(l)} = \text{HGCN}(\boldsymbol{x}^{(l-1)}), \quad l = 1, \cdots, l_G. \tag{33}$$

Finally, we take the centroid of the embeddings of all vertices to get the hyperbolic embedding $\boldsymbol{z}_G$ of the entire graph,

$$\boldsymbol{z}_G = \text{HCent}(\boldsymbol{x}^{(l_G)}). \tag{34}$$

The tree encoder is similar with the graph encoder, it encodes the junction tree to hyperbolic embedding $\boldsymbol{z}_T$ with a hyperbolic GCN of depth $l_T$. The only difference is that its input feature $\boldsymbol{x}_i$'s are one-hot vectors representing the atom clusters in the cluster vocabulary. We need to use a hyperbolic embedding layer as the first layer of the network accordingly. Altogether, the tree encoder contains

the following layers in a sequential manner:

$$
\begin{aligned}
\boldsymbol{x}_i^{(0)} &= \mathrm{HEmbed}_{d_{T_0}, d_T}(\boldsymbol{x}_i), \\
\boldsymbol{x}^{(l)} &= \mathrm{HGCN}(\boldsymbol{x}^{(l-1)}), \quad l = 1, \cdots, l_T, \\
\boldsymbol{z}_T &= \mathrm{HCent}(\boldsymbol{x}^{(l_T)}).
\end{aligned}
\tag{35}
$$

### E.2 TREE DECODER

Similar to Jin et al. (2018; 2019), we generate a junction tree $T = (V_T, E_T)$ using a tree recurrent neural network in a top-down and node-by-node fashion. The generation process resembles a depth-first traversal over the tree $T$. Staring from the root, at each time step $t$, the model makes a decision whether to continue generating a child node or backtracking to its parent node. If it decides to generate a new node, it will further predict the cluster label of the child node. It makes these decision based on the messages passed from the neighboring node. We remark that we do not use the gated recurrent unit (GRU) for message passing. The complex structure of GRU would make the training process numerically unstable for our hyperbolic neural network. We simply replace it with a hyperbolic linear layer.

**Message Passing**   Let $\tilde{E} = \{(i_1, j_1), \ldots, (i_m, j_m)\}$ denote the collection of the edges visited in a depth-first traversal over $T$, where $m = 2|E_T|$. We store a hyperbolic message $\boldsymbol{h}_{i_t, j_t}$ for each edge in $\tilde{E}$. Let $\tilde{E}_t$ be the set of the first $t$ edges in $\tilde{E}$. Suppose at time step $t$, the model visit node $i_t$ and it visits node $j_t$ at the next time step. The message $\boldsymbol{h}_{i_t, j_t}$ is updated using the node feature $\boldsymbol{x}_{i_t}$ and inward messages $\boldsymbol{h}_{k, i_t}$. We first use hyperbolic centroid to gather the inward messages to produce

$$
\boldsymbol{z}_{\mathrm{nei}} = \mathrm{HCent}(\mathrm{HLinear}_{d_T, d_T}(\{\boldsymbol{h}_{k, i_t}\}_{(k, i_t) \in \tilde{E}, k \neq j_t})),
\tag{36}
$$

and then map the tree node features to the hyperbolic space to produce

$$
\boldsymbol{z}_{\mathrm{cur}} = \mathrm{HEmbed}_{d_{T_0}, d_T}(\boldsymbol{x}_{i_t}).
\tag{37}
$$

Finally, we combine them using the Lorentz Direct Concatenation and pass them through a hyperbolic linear layer to get the message

$$
\boldsymbol{h}_{i_t, j_t} = \mathrm{HLinear}_{2 \times d_T, d_T}\left(\mathrm{HCat}(\{\boldsymbol{z}_{\mathrm{cur}}, \boldsymbol{z}_{\mathrm{nei}}\})\right).
\tag{38}
$$

**Topological Prediction**   At each time step $t$, the model makes a binary decision on whether to generate a child node, using tree embedding $\boldsymbol{z}_T$, node feature $\boldsymbol{x}_{i_t}$, and inward messages $\boldsymbol{h}_{k, i_t}$ using the following layers successively:

$$
\begin{aligned}
\boldsymbol{z}_{\mathrm{nei}} &= \mathrm{HCent}(\mathrm{HLinear}_{d_T, d_T}(\{\boldsymbol{h}_{k, i_t}\}_{(k, i_t) \in \tilde{E}})), \\
\boldsymbol{z}_{\mathrm{cur}} &= \mathrm{HEmbed}_{d_{T_0}, d_T}(\boldsymbol{x}_{i_t}), \\
\boldsymbol{z}_{\mathrm{all}} &= \mathrm{HLinear}_{3 \times d_T, d_T}\left(\mathrm{HCat}(\{\boldsymbol{z}_{\mathrm{cur}}, \boldsymbol{z}_{\mathrm{nei}}, \boldsymbol{z}_T\})\right), \\
\boldsymbol{p}_t &= \mathrm{Softmax}(\mathrm{HCDist}_{d_T, 2}(\boldsymbol{z}_{\mathrm{all}})).
\end{aligned}
\tag{39}
$$

**Label Prediction**   If a child node $j_t$ is generated, we use the tree embedding $\boldsymbol{z}_T$ and the outward message $\boldsymbol{h}_{i_t, j_t}$ to predict its label. We apply the following two layers successively:

$$
\begin{aligned}
\boldsymbol{z}_{\mathrm{all}} &= \mathrm{HLinear}_{2 \times d_T, d_T}\left(\mathrm{HCat}(\{\boldsymbol{h}_{i_t, j_t}, \boldsymbol{z}_T\})\right), \\
\boldsymbol{q}_t &= \mathrm{Softmax}(\mathrm{HCDist}_{d_T, d_{T_0}}(\boldsymbol{z}_{\mathrm{all}})).
\end{aligned}
\tag{40}
$$

The output $\boldsymbol{q}_t$ is a distribution over the label vocabulary. When $j_t$ is a root node, its parent $i_t$ is dummy and the message is padded with the origin of the hyperbolic space $\boldsymbol{h}_{i_t, j_t} = \boldsymbol{o}$.

**Training**   The topological and label prediction have two induced losses. Suppose $\hat{\boldsymbol{p}}_t, \hat{\boldsymbol{q}}_t$ are the the ground truth topological and label values, obtained by doing depth-first traversal on the real junction tree. The decoder minimizes the following cross-entropy loss:

$$
L_{\mathrm{topo}} = \sum_{t=1}^{m} L_{\mathrm{cross}}(\hat{\boldsymbol{p}}_t, \boldsymbol{p}_t), \ L_{\mathrm{label}} = \sum_{t=1}^{m} L_{\mathrm{cross}}(\hat{\boldsymbol{q}}_t, \boldsymbol{q}_t),
\tag{41}
$$

where $L_{\mathrm{cross}}$ is the cross-entropy loss. During the training phase, we use the teacher forcing strategy: after the predictions at each time step, we replace them with the ground truth. This allows the model to learn from the correct history information.

### E.3 Graph Decoder

The graph decoder assembles a molecular graph given a junction tree $\hat{T} = (\hat{V}, \hat{E})$ and graph embedding $\boldsymbol{z}_G$. Let $\mathcal{G}_i$ be the set of possible candidate subgraphs around tree node $i$, i.e. the different ways of attaching neighboring clusters to cluster $i$. We want to design a scoring function for each candidate subgraph $G_j^{(i)} \in \mathcal{G}_i$.

To this end, we first use the hyperbolic GCN and hyperbolic centroid to acquire the hyperbolic embedding $\boldsymbol{z}_{G_j^{(i)}}$ of each subgraph $G_j^{(i)}$. Specifically,

$$
\begin{aligned}
\boldsymbol{x}_v^{(0)} &= \text{E2H}_{d_{G_0}}(\boldsymbol{x}_v), \\
\boldsymbol{x}^{(l)} &= \text{HGCN}(\boldsymbol{x}^{(l-1)}), \quad l = 1, \cdots, l_G, \\
\boldsymbol{z}_{G_j^{(i)}} &= \text{HCent}(\boldsymbol{x}^{(l_G)}).
\end{aligned}
\tag{42}
$$

Then, the embedding of the subgraph is combined with the embedding of the molecular graph $\boldsymbol{z}_G$ by the Lorentz Direct Concatenation to produce

$$
\boldsymbol{z}_{\text{all}} = \text{Hlinear}_{2 \times d_G, d_G}(\text{HCat}(\{\boldsymbol{z}_{G_j^{(i)}}, \boldsymbol{z}_G\})),
\tag{43}
$$

which is then passed to the hyperbolic centroid distance layer to get a score

$$
s_j^{(i)} = \text{HCDist}_{d_G, 1}(\boldsymbol{z}_{\text{all}}) \in \mathbb{R}.
\tag{44}
$$

**Training** We define the loss for the graph decoder to be the sum of the cross-entropy losses in each $\mathcal{G}_i$. Specifically, suppose the correct subgraph is $G_c^{(i)}$,

$$
L_{\text{assm}} = \sum_i \left( s_c^{(i)} - \log \sum_{G_j^{(i)} \in \mathcal{G}_i} \exp(s_j^{(i)}) \right).
\tag{45}
$$

Similar to the tree decoder, we also use teacher forcing when training the graph decoder.

## F Detailed Settings of Experiments

### F.1 Optimization

For all the experiments, we use the Geoopt package (Kochurov et al., 2020) for Riemannian optimization. In particular, we use the Riemannian Adam function for gradient descent. We also use Geoopt for initializing the weights in all hyperbolic linear layers of our model with the wrapped normal distribution.

### F.2 Environments

All the experiments in this paper are conducted with the following environments.

- GPU: RTX 3090
- CUDA Version: 11.1
- PyTorch Version: 1.9.0
- RDKit Version: 2020.09.1.0

### F.3 Experiment Details for Toy Distribution

We describe the details for the toy distribution experiment.

### F.3.1 ARCHITECTURE DETAILS OF HYPERBOLIC GENERATIVE ADVERSARIAL NETWORK

**Generator**

- Input: points in $\mathbb{L}_K^{256}$ sampled from $\mathcal{G}(\boldsymbol{o}, \mathrm{diag}(\mathbf{1}_{256}))$
- Hyperbolic linear layers:
  - Input dimension: 256
  - Hidden dimension: 128
  - Depth: 3
  - Output dimension: 3
- Output: points in $\mathbb{L}_K^3$

**Critic**

- Input: points in $\mathbb{L}_K^3$
- Hyperbolic linear layers:
  - Input dimension: 3
  - Hidden dimension: 128
  - Depth: 3
  - Output dimension: 128
- Hyperbolic centroid distance layer: $\mathbb{L}_K^{128} \to \mathbb{R}$
- Output: score in $\mathbb{R}$

**Hyperparameters**

- Manifold curvature: $K = -1.0$
- Gradient penalty coefficient: $\lambda = 10$
- For all hyperbolic linear layers:
  - Dropout: 0.0
  - Use bias: True
- Optimizer: Riemannian Adam ($\beta_1 = 0, \beta_2 = 0.9$)
- Learning Rate: 1e-4
- Batch size: 128
- Number of epochs: 20
- Gradient penalty $\lambda$: 10

### F.4 EXPERIMENT DETAILS FOR MNIST GENERATION

We describe the detailed architecture for the MNIST Generation experiment.

### F.4.1 ARCHITECTURE DETAILS OF AUTO-ENCODER

**Encoder**

- Input: MNIST image with dimension $(28 \times 28)$
- Convolutional Neural Network Encoder
  - Convolutional layer
    * Input channel: 1
    * Output channel: 8
    * Kernel Size: 3
    * Stride: 2
    * Padding: 1

- Leaky ReLU (0.2)
- Convolutional layer
  * Input channel: 8
  * Output channel: 16
  * Kernel Size: 3
  * Stride: 2
  * Padding: 1
- Batch normalization layer (16)
- Leaky ReLU (0.2)
- Convolutional layer
  * Input channel: 16
  * Output channel: 32
  * Kernel Size: 3
  * Stride: 2
  * Padding: 0
- Batch normalization layer (32)
- Leaky ReLU (0.2)
- Convolutional layer
  * Input channel: 32
  * Output channel: 64
  * Kernel Size: 3
  * Stride: 2
  * Padding: 0

- Map to hyperbolic space: $\mathbb{R}^{64} \to \mathbb{L}_K^{64}$
- Hyperbolic linear layers:
  - Input dimension: 64
  - Hidden dimension: 64
  - Depth: 3
  - Output dimension: 64
- Output: hyperbolic embeddings in $\mathbb{L}_K^{64}$

**Decoder**

- Input: hyperbolic embeddings in $\mathbb{L}_K^{64}$
- Hyperbolic linear layers:
  - Input dimension: 64
  - Hidden dimension: 64
  - Depth: 3
  - Output dimension: 64
- Map to Euclidean space: $\mathbb{L}_K^{64} \to \mathbb{R}^{64}$
- Transposed Convolutional Neural Network Decoder
  - Transposed Convolutional layer
    * Input channel: 64
    * Output channel: 64
    * Kernel Size: 3
    * Stride: 1
    * Padding: 0
    * Output Padding: 0
  - Batch normalization layer (64)
  - Leaky ReLU (0.2)
  - Transposed Convolutional layer

- * Input channel: 64
        * Output channel: 32
        * Kernel Size: 3
        * Stride: 2
        * Padding: 0
        * Output Padding: 0
    – Batch normalization layer (32)
    – Leaky ReLU (0.2)
    – Transposed Convolutional layer
        * Input channel: 32
        * Output channel: 16
        * Kernel Size: 3
        * Stride: 2
        * Padding: 1
        * Output Padding: 1
    – Batch normalization layer (16)
    – Leaky ReLU (0.2)
    – Transposed Convolutional layer
        * Input channel: 16
        * Output channel: 1
        * Kernel Size: 3
        * Stride: 2
        * Padding: 1
        * Output Padding: 1
- Output: Reconstructed MNIST image with dimension $(28 \times 28)$

**Hyperparameters**

- Manifold curvature: $K = -1.0$
- For all hyperbolic linear layers:
    – Dropout: 0.0
    – Use bias: True
- Optimizer: Riemannian Adam ($\beta_1 = 0.9, \beta_2 = 0.9$)
- Learning Rate: 1e-4
- Batch size: 32
- Number of epochs: 20

### F.4.2 ARCHITECTURE DETAILS OF HYPERBOLIC GENERATIVE ADVERSARIAL NETWORK

**Generator**

- Input: points in $\mathbb{L}_K^{128}$ sampled from $\mathcal{G}(\boldsymbol{o}, \mathrm{diag}(\mathbf{1}_{128}))$
- Hyperbolic linear layers:
    – Input dimension: 128
    – Hidden dimension: 64
    – Depth: 3
    – Output dimension: 64
- Output: points in $\mathbb{L}_K^{64}$

**Critic**

- Input: points in $\mathbb{L}_K^{64}$
- Hyperbolic linear layers:
  - Input dimension: 3
  - Hidden dimension: 64
  - Depth: 3
  - Output dimension: 64
- Hyperbolic centroid distance layer: $\mathbb{L}_K^{64} \to \mathbb{R}$
- Output: score in $\mathbb{R}$

**Hyperparameters**

- Manifold curvature: $K = -1.0$
- Gradient penalty coefficient: $\lambda = 10$
- For all hyperbolic linear layers:
  - Dropout: 0.0
  - Use bias: True
- Optimizer: Riemannian Adam ($\beta_1 = 0, \beta_2 = 0.9$)
- Learning Rate: 1e-4
- Batch size: 64
- Number of epochs: 20
- Gradient penalty $\lambda$: 10

### F.5  EXPERIMENT DETAILS FOR MOLECULAR GENERATION

We describe the detailed architecture and settings for the molecular generation experiments.

#### F.5.1  ARCHITECTURE DETAILS OF HYPERBOLIC JUNCTION TREE ENCODER-DECODER

**Graph Encoder**

- Input: graph node features in $\mathbb{R}^{35}$
- Map features to hyperbolic space: $\mathbb{R}^{35} \to \mathbb{L}_K^{35}$
- Hyperbolic GCN layers:
  - Input dimension: 35
  - Hidden dimension: 256
  - Depth: 4
  - Output dimension: 256
- Hyperbolic centroid on all vertices
- Output: graph embedding in $\mathbb{L}_K^{256}$

**Tree Encoder**

- Input: junction tree features in $\mathbb{R}^{828}$
- Hyperbolic embedding layer: $\mathbb{R}^{828} \to \mathbb{L}_K^{256}$
- Hyperbolic GCN layers:
  - Input dimension: 256
  - Hidden dimension: 256
  - Depth: 4
  - Output dimension: 256
- Hyperbolic centroid on all vertices
- Output: tree embedding in $\mathbb{L}_K^{256}$

**Tree Decoder**

- Input: tree embedding in $\mathbb{L}_K^{256}$
- Message passing RNN:
  - Input: node feature of current tree node, inward messages
  - Hyperbolic linear layer on inward messages: $\mathbb{L}_K^{256} \to \mathbb{L}_K^{256}$
  - Hyperbolic centroid on inward messages
  - Hyperbolic embedding layer on node feature: $\mathbb{R}^{828} \to \mathbb{L}_K^{256}$
  - Lorentz Direct concatenation on node feature and inward message: $\mathbb{L}_K^{256} \to \mathbb{L}_K^{512}$
  - Hyperbolic linear layer: $\mathbb{L}_K^{512} \to \mathbb{L}_K^{256}$
  - Output dimension: 256
- Topological Prediction:
  - Input: tree embedding, node feature of current tree node, inward messages
  - Hyperbolic linear layer on inward messages: $\mathbb{L}_K^{256} \to \mathbb{L}_K^{256}$
  - Hyperbolic centroid on inward messages
  - Hyperbolic embedding layer on tree feature: $\mathbb{R}^{828} \to \mathbb{L}_K^{256}$
  - Lorentz Direct concatenation on node feature, inward message, and tree embedding: $\mathbb{L}_K^{256} \to \mathbb{L}_K^{768}$
  - Hyperbolic linear layer: $\mathbb{L}_K^{768} \to \mathbb{L}_K^{256}$
  - Hyperbolic centroid distance layer: $\mathbb{L}_K^{256} \to \mathbb{R}^2$
  - Softmax on output
  - Output dimension: 2
- Label Prediction:
  - Input: tree embedding, outward messages
  - Lorentz Direct concatenation on outward message, and tree feature: $\mathbb{L}_K^{256} \to \mathbb{L}_K^{512}$
  - Hyperbolic linear layer: $\mathbb{L}_K^{512} \to \mathbb{L}_K^{256}$
  - Hyperbolic centroid distance layer: $\mathbb{L}_K^{256} \to \mathbb{R}^{828}$
  - Softmax on output
  - Output dimension: 828
- Output: junction tree

**Graph Decoder**

- Input: junction tree, tree message, and graph embedding
- Construction candidate subgraphs
- Hyperbolic graph convolution layers on all subgraphs:
  - Input dimension: 256
  - Hidden dimension: 256
  - Depth: 4
  - Output dimension: 256
- Hyperbolic centroid on vertices of all subgraphs
- Lorentz Direct concatenation on subgraph embedding and graph embedding: $\mathbb{L}_K^{256} \to \mathbb{L}_K^{512}$
- Hyperbolic linear layer: $\mathbb{L}_K^{512} \to \mathbb{L}_K^{256}$
- Hyperbolic centroid distance layer: $\mathbb{L}_K^{256} \to \mathbb{R}$
- Use subgraph score to construct molecular graph
- Output: molecular graph

**Hyperparameters**

- Manifold curvature: $K = -1.0$
- For all hyperbolic linear layers:
  - Dropout: 0.0
  - Use bias: True
- Optimizer: Riemannian Adam ($\beta_1 = 0.0, \beta_2 = 0.999$)
- Learning rate: 5e-4
- Learning rate scheduler: StepLR (step = 20000, $\gamma = 0.5$)
- Batch size: 32
- Number of epochs: 20

### F.5.2 ARCHITECTURE DETAILS OF HYPERBOLIC GENERATIVE ADVERSARIAL NETWORK

**Generator**

- Input: points sampled from wrapped normal distribution $\mathcal{G}(\boldsymbol{o}, \mathrm{diag}(\boldsymbol{1}_{128}))$ in $\mathbb{L}_K^{128}$
- Hyperbolic linear layers for graph embedding:
  - Input dimension: 128
  - Hidden dimension: 256
  - Depth: 3
  - Output dimension: 256
- Hyperbolic linear layers for tree embedding:
  - Input dimension: 128
  - Hidden dimension: 256
  - Depth: 3
  - Output dimension: 256
- Output: graph embedding and tree embedding in $\mathbb{L}_K^{128}$

**Critic**

- Input: graph embedding and tree embedding in $\mathbb{L}_K^{128}$
- Hyperbolic linear layers for graph embedding:
  - Input dimension: 256
  - Hidden dimension: 256
  - Depth: 2
  - Output dimension: 256
- Hyperbolic linear layers for tree embedding:
  - Input dimension: 256
  - Hidden dimension: 256
  - Depth: 2
  - Output dimension: 256
- Lorentz Direct concatenation on graph embedding and tree embedding: $\mathbb{L}_K^{256} \rightarrow \mathbb{L}_K^{512}$
- Hyperbolic linear layer: $\mathbb{L}_K^{512} \rightarrow \mathbb{L}_K^{256}$
- Hyperbolic centroid distance layer: $\mathbb{L}_K^{256} \rightarrow \mathbb{R}$
- Output: score in $\mathbb{R}$

**Hyperparameters**

- Manifold curvature: $K = -1.0$
- Gradient penalty coefficient: $\lambda = 10$
- For all hyperbolic linear layers:
    - Dropout: 0.1
    - Use bias: True
- Optimizer: Riemannian Adam ($\beta_1 = 0, \beta_2 = 0.9$)
- Learning Rate: 1e-4
- Batch size: 64
- Number of epochs: 20
- Gradient penalty $\lambda$: 10

### F.6  EXPERIMENT DETAILS FOR TREE GENERATION

We describe the detailed architecture and settings for the tree generation experiments.

### F.6.1  ARCHITECTURE DETAILS OF HYPERBOLIC TREE ENCODER-DECODER

**Tree Encoder**

- Input: tree
- Hyperbolic GCN layers:
    - Input dimension: 1
    - Hidden dimension: 32
    - Depth: 2
    - Output dimension: 32
- Hyperbolic centroid on all vertices
- Output: tree embedding in $\mathbb{L}_K^{32}$

**Tree Decoder**

- Input: tree embedding in $\mathbb{L}_K^{32}$
- Message passing RNN:
    - Input: node feature of current tree node, inward messages
    - Hyperbolic linear layer on inward messages: $\mathbb{L}_K^{32} \to \mathbb{L}_K^{32}$
    - Hyperbolic centroid on inward messages
    - Hyperbolic linear layer: $\mathbb{L}_K^{32} \to \mathbb{L}_K^{32}$
    - Output dimension: 32
- Topological Prediction:
    - Input: tree embedding, inward messages
    - Hyperbolic linear layer on inward messages: $\mathbb{L}_K^{32} \to \mathbb{L}_K^{32}$
    - Hyperbolic centroid on inward messages
    - Lorentz Direct concatenation on inward message and tree embedding: $\mathbb{L}_K^{32} \to \mathbb{L}_K^{64}$
    - Hyperbolic linear layer: $\mathbb{L}_K^{64} \to \mathbb{L}_K^{32}$
    - Hyperbolic centroid distance layer: $\mathbb{L}_K^{32} \to \mathbb{R}^2$
    - Softmax on output
    - Output dimension: 2
- Output: tree

**Hyperparameters**

- Manifold curvature: $K = -1.0$
- For all hyperbolic linear layers:
    - Dropout: 0.0
    - Use bias: True
- Optimizer: Riemannian Adam ($\beta_1 = 0.0, \beta_2 = 0.999$)
- Learning rate: 5e-3
- Learning rate scheduler: StepLR (step = 20000, $\gamma = 0.5$)
- Batch size: 32
- Number of epochs: 20

### F.6.2 ARCHITECTURE DETAILS OF HYPERBOLIC GENERATIVE ADVERSARIAL NETWORK

**Generator**

- Input: points sampled from wrapped normal distribution $\mathcal{G}(\boldsymbol{o}, \mathrm{diag}(\mathbf{1}_{16}))$ in $\mathbb{L}_K^{16}$
- Hyperbolic linear layers for tree embedding:
    - Input dimension: 16
    - Hidden dimension: 32
    - Depth: 2
    - Output dimension: 32
- Output: tree embedding in $\mathbb{L}_K^{32}$

**Critic**

- Input: tree embedding in $\mathbb{L}_K^{32}$
- Hyperbolic linear layers for tree embedding:
    - Input dimension: 32
    - Hidden dimension: 32
    - Depth: 2
    - Output dimension: 32
- Hyperbolic centroid distance layer: $\mathbb{L}_K^{32} \to \mathbb{R}$
- Output: score in $\mathbb{R}$

**Hyperparameters**

- Manifold curvature: $K = -1.0$
- Gradient penalty coefficient: $\lambda = 10$
- For all hyperbolic linear layers:
    - Dropout: 0.1
    - Use bias: True
- Optimizer: Riemannian Adam ($\beta_1 = 0, \beta_2 = 0.9$)
- Learning Rate: 1e-4
- Batch size: 64
- Number of epochs: 20
- Gradient penalty $\lambda$: 10

# G  MOLECULE EXAMPLES

We show a subset of molecule examples generated by HAEGAN.

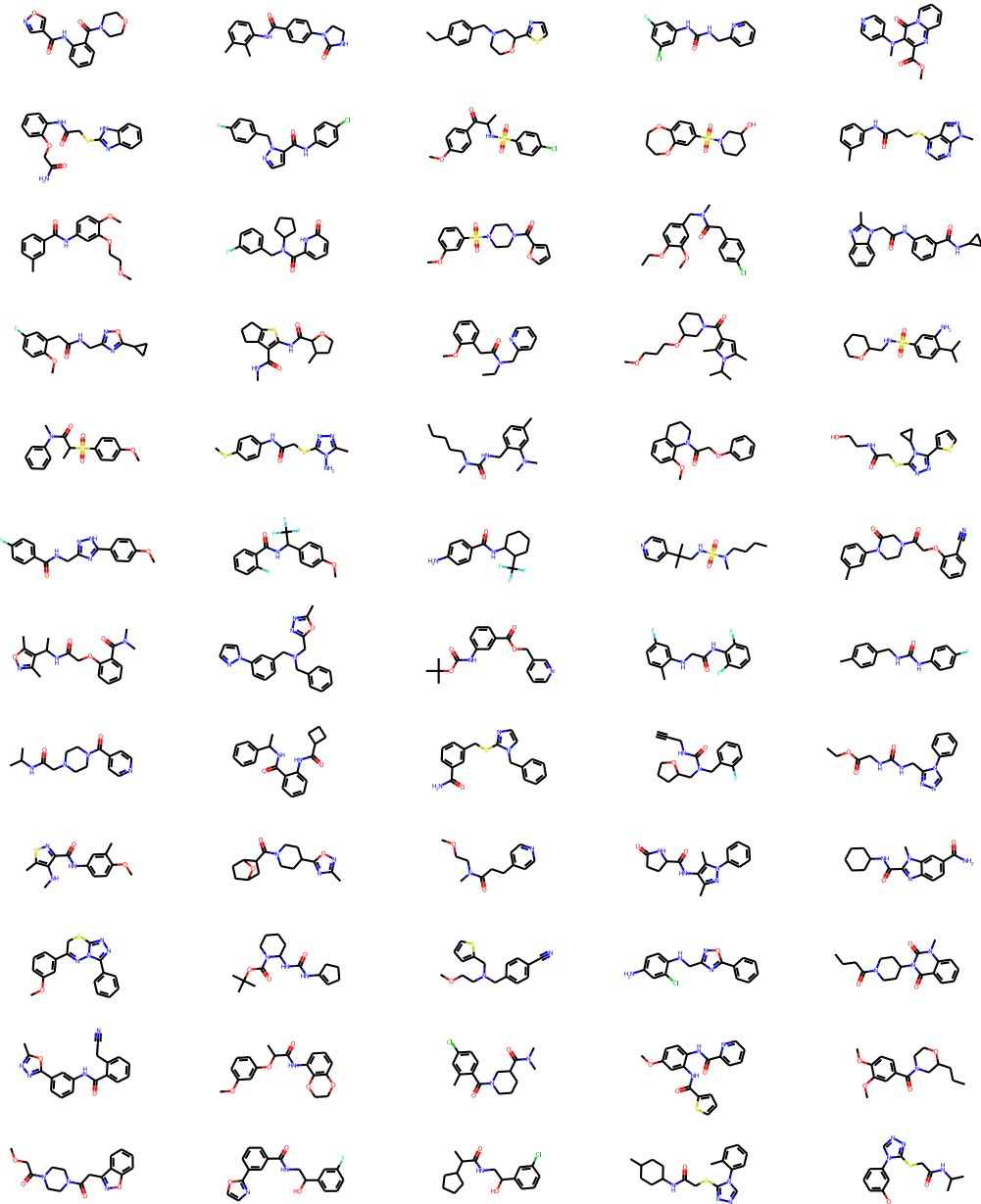

Figure 12: Molecule examples generated by HAEGAN.

