# OpenReview forum: "Autoencoding Hyperbolic Representation for Adversarial Generation"
_ICLR.cc/2023/Conference — Submitted to ICLR 2023_

### Official Review · Reviewer_PLXj · 2022-10-24

**Confidence:** 4
**Correctness:** 3
**Technical Novelty And Significance:** 3
**Empirical Novelty And Significance:** 3
**Recommendation:** 6

**Clarity, Quality, Novelty And Reproducibility:**

The paper follows clear logic, easy to follow. Lack of enough novelty. Some statements/claims need to be changed.

**Strength And Weaknesses:**

Strength. -- The proposed framework follows clear logic and easy to follow, mostly used the layer definition and function in [Chen et al. 2021] fully hyperbolic neural network. The authors also propose a straightforward version for concatenating/split vectors in the Lorentz model, which empirically shows stable gradients during optimization. The comparison of Lorentz Direct Concatenation and Lorentz Tangent Concatenation looks interesting and helpful.

-- The authors evaluate the proposed HAEGAN on learning molecules structure and compare against several baselines using structure metrics.

-- The improvement on learning structure of molecules seems to be significant when compared with various baselines.

Weakness. -- Novelty of the model. Apart from the proposed concatenating/split, all other layers were proposed in prior works, particularly from [Chen et al. 2021], the whole framework looks to me a combination of existing pieces into a hyperbolic version of common GAN.

-- The authors claim that the stable training of the model is guaranteed. However, the numerical stable property is only evaluated for the concatenating/split operation empirically. The numerical stable hyperbolic centroid distance layer is proposed in early work. There is no theory to guarantee the stable training. Particularly, large gradients into exp/log map is one side of the numerical instability, the other side of the numerical instability comes from float representation of hyperbolic space, a line of work such as [Yu et al. 2019], which happens when the hyperbolic point becomes far from the origin. Please modify the claim in the paper in an appropriate way.

Suggestion and Question:
-- Instead of putting 5 rows of NaN in table 1, you can just illustrate them in plain text. Though still it looks surprising to me that all hyperbolic baselines produce NaN, can you find a hyperbolic baseline without NaN, i.e., with less epochs or small graph just for a comparison?

* [Chen et al. 2021], Fully hyperbolic neural networks.
* [Yu et al. 2019], Numerically Accurate Hyperbolic Embeddings Using Tiling-Based Models

**Summary Of The Paper:**

A proposal of hyperbolic generative models is provided in the paper based on some existing structures in the literature, a stable training/optimization is observed empirically. The authors adopt the framework to learning molecule structure with the goal to learn the structural information.

**Summary Of The Review:**

The paper proposes a new framework of hyperbolic GAN and adopt it in a new task, learning the structure of molecules. The proposed model outperforms Euclidean baselines, where hyperbolic baselines are not available for comparison. However, the paper is lack of enough novelty, as most used layers are proposed before.

---

> ### Author Response · Authors · 2022-11-14
> **Response to Reviewer PLXj**
>
> Thank you for your time and effort in reviewing our paper. We respond to the weaknesses and questions you mentioned in your review as follows.
>
> ### Weaknesses:
>
> > Novelty of the model. Apart from ... into a hyperbolic version of common GAN.
>
> We made more effort than simply combining existing work. First, the Euclidean WGAN-GP could not be directly adopted into the hyperbolic space, we proposed a novel gradient penalty for Hyperbolic WGAN and proved its correctness (Proposition 3.1). Then, we modified the junction tree graph generation framework into hyperbolic operations (Appendix E). Furthermore, directly incorporating the tree and graph AE into hyperbolic WGAN or HVAE would produce numerically unstable results and we proposed the novel overall framework to ensure stable training, where the AE plays an important role.
> In the revised version, we have also added more theoretical analysis of Lorentz concatenations (Theorem 4.1) in addition to the analysis we had in Appendix D.
>
>
> >  The authors claim that the stable training of the model is guaranteed. However ... modify the claim in the paper in an appropriate way.
>
> We have added theoretical analysis on the numerical stability of Lorentz direct concatenation and instability of Lorentz tangent concatenation (Theorem 4.1).
> Thank you for pointing out the two sides (gradient explosion and float precision). We did not intend to claim that the overall network has a guaranteed stability. Of course, if we look at a single fully hyperbolic linear layer or a single direct concatenation layer, there would be no problem with both causes of instability since there are no exponential or logrithmic maps. We have modified our claim about stable training following your suggestion and added the appropriate citations in $\S$ 4.1.
>
>
> ### Suggestion and Question:
>
> >  Instead of putting 5 rows of NaN in table 1, you can just illustrate them in plain text.
>
> Thank you for the suggestion, we have modified the table in the revised manuscript.
>
> > Though still it looks surprising to me that all hyperbolic baselines produce NaN, can you find a hyperbolic baseline without NaN, i.e., with less epochs or small graph just for a comparison?
>
> We have added a new random tree generation experiment in $\S$ 5.1 where not all hyperbolic baseline produce NaN results (still, the hyperbolic VAEs are unstable). In this experiment, HAEGAN outperforms the hyperbolic baselines in all metrics except for one.
> Our MNIST generation experiment (served as a sanity check since there is no clear tree or hierarchical structure in MNIST), reported in Appendix C, also quantitatively compares HAEGAN with other hyperbolic baselines.

---

### Official Review · Reviewer_V3p6 · 2022-10-25

**Confidence:** 3
**Correctness:** 3
**Technical Novelty And Significance:** 3
**Empirical Novelty And Significance:** 3
**Recommendation:** 6

**Clarity, Quality, Novelty And Reproducibility:**

**Clarity**
--

The paper is very well written and easy to follow.

**Novelty and Significance**
---

While the architecture is novel, the motivation behind it is not the clearest. The results produced do however seem significant.

**Reproducible**
---

The paper seems to be reproducible.

**Questions**
---

1) The results for the variants HVAE-w, HVAE-r, and HAEGAN-H surprise me. (The other seem reasonable). Specifically, the VAE ones. VAEs are  fairly stable to train (to my knowledge) and these use the fully hyperbolic layers. Hence I would imagine that while this model might not perform well it should train and produce non-NAN results. The HAEGAN-H is less surprising but is still surprising. Maybe this is an issue related to learning rate and the learning rate scheduler?


**Strength And Weaknesses:**

**Strengths**
--

1) I think Proposition 3.1 is a strength. WGANs have improved GAN stability and being able to use a similar formulation (beyond the architecture mentioned here) is interesting and significant.
2) The method seems to perform well and produce valid and diverse molecules.
3) The paper is very well written. It is very easy to read and follow. All details are present and it should be easily reproducible.
4) The ablation study is well conducted.

**Weakness**
---

1) The motivation for the architecture is not quite clear. The authors, mention that training Hyperbolic Neural Networks is unstable and hence GANs can't model complex distributions. While I tend to believe this for the older methods that use the Tangent space, I however, do not know if this is true for the newer method that uses Lorentz rotations and boosts. Hence evidence of this via experimentation would be appreciated.

Second, this issues rather being resolved is pushed to a different part and then not discussed. In a sense, the method still learns a map from wrapped gaussians to the data distribution (GAN generator + Autoencoder decoder). Hence the autoencoder must now figure out how to map from a complex distribution to a simple distribution and then back to the complex distribution. Maybe because the training of autoencoders is more stable than GANs this is okay.

Finally, the representation learned by the autoencoder is supposed to ``simpler'' than the original representation. But lower dimensional representations of data need not be simpler than higher dimensional representations. More discussion on this is needed.

2) While the paper in general does a good literature survey, the paper has a section 2.3 which is on Embedding form Euclidean to Hyperbolic Spaces, which misses crucial literature. The paper claims it is unavoidable to use the exponential or logarithmic maps to represent the data hyperbolic space. However, this is not true. There are many embedding techniques such as Nickel and Kiela NeurIPS 2017, Nickel and Kiela ICML 2018, Sala, De Sa, Gu and Re ICML 2018, Sonthalia and Gilbert NeurIPS 2020. In particular, Sala et al 2018 and Sonthalia and Gilbert 2020 do not use the exponential or logarithmic maps. (This is a more minor comment.)

**Summary Of The Paper:**

The paper presents a new architecture for generative modeling that makes use of hyperbolic geometry. Specifically, the builds on Chen et al 2022 to propose a generative framework that combines both Autoencoders and GANs. Additionally, the paper also a presents a new method for splitting and concatenating vectors in the Lorentz manifold. The paper adapts the $W$-GAN framework and proves a similar proposition for the characterization of the Wasserstein Distance.

The paper then tests these pieces out individually - generating images from MNIST, testing their concatenation method and then finally generating molecules using their method.

**Summary Of The Review:**

In summary, I think this is a well written paper with interesting results. However, some parts could be better explained.

---

> ### Author Response · Authors · 2022-11-14
> **Response to Reviewer V3p6**
>
> Thank you for your time and effort in reviewing our paper. We respond to the weaknesses and questions you mentioned in your review as follows.
>
> ### Weakness 1:
>
> We believe your concern is about $\S$ 3.2, where the purpose of adding the AE is rather unclear. We have modified it in the revision to clarify that the purpose of using the AE + GAN framework is to ensure numerical stability. The training of GANs is known to be unstable. If we only use a GAN, then we need all the hyperbolic layers in the generator and the critic. However, *when we incorporate complex network architectures into the generator and critic, it will lead to numerical instability*. Learning the hyperbolic representation with AE ensures that we can use a relatively simpler GAN while not losing expressivity. To answer your concern, we do not necessarily need the AE embedding to be simple, but only require that it could be learned by a hyperbolic WGAN.
>
> We are now demonstrating the instability of hyperbolic WGAN by adding an ablation in the molecular generation experiment, where we train an end-to-end hyperbolic WGAN (HGAN) with the graph and tree decoder as the generator, and the graph and tree encoder as the critic. HGAN also has produced numerically unstable results. Specifically, its loss diverged quickly and turned into NaN after a few epochs.
>
> We have added a new random tree generation experiment in $\S$ 5.1 in which we also tested a simpler HGAN.
> We note that the training process of this HGAN is quite sensitive to the learning rate, which also indicates the instability of hyperbolic GANs.
>
> ### Weakness 2:
>
> Thank you for careful reading. We are now mentioning these references in Appendix A.3 (we moved some background to the appendix following the suggestion of Reviewer 6PpY). We did not intend to mention E2H as the only embedding method, but as a simple operation that can be incorporated in HAEGAN. Please see the revised manuscript.
>
> ### Questions:
>
> The problem with the hyperbolic VAE models is that the ELBO formulations \[1\] for wrapped normal distribution and Riemannian normal distribution are extremely complex. When incorporating the ELBO loss with the complicated tree and graph AE, the model became numerically unstable. As for HAEGAN-H, the instability comes from the stacking of logarithmic and exponential maps, which is similar to HAEGAN-T. We had tried altering the learning rate to be extremely small, but the models would still produce NaN loss.
>
> \[1\] Mathieu, E., Le Lan, C., Maddison, C. J., Tomioka, R., & Teh, Y. W. (2019). Continuous hierarchical representations with poincaré variational auto-encoders. *Advances in neural information processing systems*, 32.

---

> > ### Comment · Reviewer_V3p6 · 2022-11-29
> > **Thank you for addressing my concerns.**
> >
> > Thank you. I still think the paper is worth accepting.

---

### Official Review · Reviewer_6PpY · 2022-10-26

**Confidence:** 3
**Correctness:** 3
**Technical Novelty And Significance:** 2
**Empirical Novelty And Significance:** 2
**Recommendation:** 3

**Clarity, Quality, Novelty And Reproducibility:**

-The organization and clarity of the paper need improvement.
-The proposed technique exhibit certain novelty.
-Code is not provided.



**Strength And Weaknesses:**

Pros:
1. This paper designs a model that can guarantee stable training of hyperbolic neural networks.
2. In the experiments, different metrics have been utilized to evaluate the proposed method. There are some interesting experimental results in this paper.

Cons:

1. It seems that this work simply combines the hyperbolic neural networks with autoencoder and generative adversarial networks. I think the authors have to clarify the novelty and contributions of this work more clearly.

2. It is not clear how the proposed model can guarantee stable training. A further explanation/clarification is necessary.

3. The organization of this paper is somewhat confusing. Lots of the contents in this paper are utilized to introduce the background (e.g., Section 2, Section 4.1, Section 6). I think the authors should put more effort on explaining the proposed model. Maybe some background introduction can be moved into Appendix. The organization and presentation in this paper should be modified.

4. The scalability of the proposed technique should be discussed.

5. The experiments are limited, the authors should carry out experiments on one or two more datasets.

6. This work lacks complexity analysis, the authors should analyze the complexity (e.g., time complexity) of the proposed method and compare it to state-of-the-art.

7. Many notations have been used in this paper, I suggest the authors could try to make a table to explain all the notations in the Appendix.


**Summary Of The Paper:**

This work proposes a new network that combines autoencoder and generative adversarial network to guarantee stable training of hyperbolic neural networks. Experiments show that the proposed method can generate complex data, and there are some interesting results in the experiments. Nevertheless, there are still some issues that need to be resolved.

**Summary Of The Review:**

Overall, I still have some concerns about the organization and clarify the proposed technique. The experiments are also limited.

---

> ### Author Response · Authors · 2022-11-14
> **Response to Reviewer 6PpY**
>
> Thank you for your time and effort in reviewing our paper. We respond to the cons you mentioned in your review as follows.
>
> > It seems that this work simply combines the hyperbolic neural networks with autoencoder and generative adversarial networks. I think the authors have to clarify the novelty and contributions of this work more clearly.
>
> The contribution of this paper was presented at the end of $\S$ 1 with bulletin points. This work is not a simple combination of hyperbolic neural networks with AE and GAN. First, the overall framework is novel (as shown in the experiments, a simple HGAN or HVAE may suffer from instability). Second, the Wasserstein GAN formulation is validated in the hyperbolic space, which is novel (Proposition 3.1). Third, the concatenation layer is a contribution, which is an important component for stability (we have added theoretical support). It is also nontrivial to construct the HAEGAN for molecular generation (See Appendix E).
>
> > It is not clear how the proposed model can guarantee stable training. A further explanation/clarification is necessary.
>
> We made the following effort to ensure stable training:
> - We proposed the Lorentz Direct Concatenation, which proves to be more stable than other concatenation methods, both empirically and theoretically (see $\S$ 4.2, particularly the newly added Theorem 4.1, and much more detailed analysis in Appendix D).
> - We validated the WGAN-GP framework in hyperbolic space, whose training is more stable than plain GAN.
> - We designed the AE + GAN framework because training AE is more stable than GAN with many cascaded hyperbolic layers.
> - We utilized the fully hyperbolic linear layer, which is more stable than the tangent linear layer.
>
> > The organization of this paper is somewhat confusing. ... The organization and presentation in this paper should be modified.
>
> On one hand, we had felt that the background contained in $\S$ 2 and $\S$ 6 provide useful information for reader not familiar with hyperbolic neural networks. In particular, $\S$ 2 contains a description of notations. On the other hand, $\S$ 4.1 is not background, but the motivation and definition of the concatenation layer that we defined.
>
> Following your suggestion, we have simplified $\S$ 2 and moved $\S$ 6 to the appendix.
>
> > The scalability of the proposed technique should be discussed.
>
> Our proposed method, particularly the concatenation layer, contains fully hyperbolic operations that do not take exponential and logarithmic maps. Therefore, by avoiding using the exponential function, our method achieves GPU scalability (see \[1\] for the same motivation but a different approach). We have added the discussion in $\S$ 4.2.
>
> \[1\] Choudhary, N., & Reddy, C. K. (2022). Towards Scalable Hyperbolic Neural Networks using Taylor Series Approximations. *arXiv preprint arXiv:2206.03610*.
>
> > The experiments are limited, the authors should carry out experiments on one or two more datasets.
>
> We have added a new random tree generation experiment in $\S$ 5.1, which shows both effectiveness and efficiency of our approach.
>
> > This work lacks complexity analysis, the authors should analyze the complexity (e.g., time complexity) of the proposed method and compare it to state-of-the-art.
>
> Our method, in particular the direct concatenation, avoids the complex exponential and logarithmic maps, which lowers the complexity. We have added the discussion in $\S$ 4.2.
> In addition, to validate the efficiency, we have recorded the runtime of different methods in the random tree generation experiment, which is reported in Table 1.
>
> > Many notations have been used in this paper, I suggest the authors could try to make a table to explain all the notations in the Appendix.
>
> The notations were summarized in $\S$ 2.1 with a paragraph "Notation". For readers unfamiliar with hyperbolic operations, simply listing the notations in a table is not sufficient. That was why we included a somehow detailed background section. Since we have rearranged the background materials, the notations should be clear following Appendices A.1-A.3.

---

### Author Response · Authors · 2022-11-14
**Response to All Reviewers**

We thank all the reviewers for their valuable comments and detailed constructive suggestions.

We are encouraged that the reviewers find our paper well-written and easy to follow (R2 [6PpY], R3 [PLXj]). We are pleased that R1 [6PpY] recognizes our effort in making the model numerically stable, R2 finds our contribution on the hyperbolic WGAN formulation "interesting and significant", and R3 finds our hyperbolic concatenation "interesting and helpful". We are also glad that the reviewers find our molecular generation experiment produces "interesting experimental results" (R1), generates "valid and diverse molecules" (R2), and the improvements in structure-related metrics are significant (R3) when compared with various baselines (R3) and well-conducted ablations (R2).

We provide point-to-point responses to each review in detail. We have also revised our manuscript according to some suggestions or in correspondence to some misunderstanding. We are submitting a revised version of the paper. Also, in the supplementary material, there is a pdf file that highlights the changes in blue color (when a whole section is new, we only color the section heading). We summarize the changes as follows (the section numbers correspond to the revised version):

- major changes:
1. We have added theoretical analysis on the numerical stability of Lorentz direct concatenation and instability of Lorentz tangent concatenation (Theorem 4.1, with a proof in Appendix D.1 and simple validation in Appendix D.2).
2. We have added a new random tree generation experiment in $\S$ 5.1 to further illustrate the effectiveness of our model. In particular, not all hyperbolic baseline methods produce NaN results and we are able to show the state-of-the-art performance of our model in this task.

- minor changes:
3. We have simplified $\S$ 2 and moved some background and related works (original $\S$ 6) to the appendix (Appendix A.2-A.4).
4. We have made several minor changes to the writing for clarification. We also have changes corresponding to some reviews, which we will refer to in our point-to-point response.

(update) We have released our code as part of the supplementary material.

---

### Decision · Program_Chairs · 2023-01-20

**Decision:**

Reject

**Justification For Why Not Higher Score:**

Empirical results are only shown for small/toy datasets. While the molecular generation experiments are interesting, the approach should ideally be tested on at least another non-toy domain.

**Justification For Why Not Lower Score:**

N/A

**Metareview: Summary, Strengths And Weaknesses:**

This paper combines hyperbolic representations with GANs to generate data with hierarchical/geometric structure. The strengths of the paper are: theoretically showing that WGAN-GP extends to hyperbolic spaces and experiments on de novo molecular generation. The main weakness of the paper is that it is mostly tested on small/toy datasets.

**Summary Of Ac-Reviewer Meeting:**

Here were some points that were discussed in the AC-reviewer meeting.

- Potential concerns about novelty (i.e., "just" an application of Hyperbolic NNs to GANs)
- Scalability might be an issue
- Data: experiments on molecules was interesting, but the other datasets seemed mostly toy-ish/synthetic.
- Some theory that shows that GAN theory generalizes to hyperbolic spaces was nice.